# Misadjustment of diurnal expression of core temperature and locomotor activity in lactating rabbits associated with maternal over-nutrition before and during pregnancy

Erika Navarrete[1], Andrea Díaz-Villaseñor[1], Georgina Díaz[1], Ana María Salazar[1], Rodrigo Montúfar-Chaveznava[2], Patricia Ostrosky-Wegman[1], Ivette Caldelas[1]*

1 Instituto de Investigaciones Biomédicas, Universidad Nacional Autónoma de México, Ciudad de México, México, 2 Facultad de Ingeniería, Universidad Nacional Autónoma de México, Ciudad de México, México

* caldelas@biomedicas.unam.mx

**Data Availability Statement:** All relevant data are within the paper and its Supporting Information files.

## Abstract

Metabolic parameters ranging from circulating nutrient levels and substrate utilization to energy expenditure and thermogenesis are temporally modulated by the circadian timing system. During critical embryonic developmental periods, maternal over-nutrition could alter key elements in different tissues associated with the generation of circadian rhythmicity, compromising normal rhythmicity development. To address this issue, we determine whether maternal over-nutrition leads to alterations in the development of circadian rhythmicity at physiological and behavioral levels in the offspring. For this, female rabbits were fed a standard diet (SD) or high-fat and carbohydrate diet (HFCD) before mating and during gestation. Core body temperature and gross locomotor activity were continuously recorded in newborn rabbits, daily measurements of body weight and the amount of milk ingested was carried out. At the end of lactation, tissue samples, including brown adipose tissue (BAT) and white adipose tissue (WAT), were obtained for determining the expression of uncoupling protein-1 (UCP1) and cell death-inducing DNA fragmentation factor-like effector A (CIDEA) genes. HFCD pups exhibited conspicuous differences in the development of the daily rhythm of temperature and locomotor activity compared to the SD pups, including a significant increase in the daily mean core temperature, changes in the time when temperature or activity remains above the average, shifts in the acrophase, decrease in the duration and intensity of the anticipatory rise previous to nursing, and changes in frequency of the rhythms. HFCD pups exhibited a significant increase in BAT thermogenesis markers, and a decrease of these markers in WAT, indicating more heat generation by brown adipocytes and alterations in the browning process. These results indicate that maternal over-nutrition alters offspring homeostatic and chronostatic regulation at the physiological and behavioral levels. Further studies are needed to determine whether these alterations are associated with the changes in the organization of the circadian system of the progeny.

**Funding:** IC received the CONACyT Fronteras de la Ciencia grant (398), Programa de Apoyo a Proyectos de Investigación e Innovación Tecnológica grant (PAPIIT IN212516 and PAPIIT IN211219). IC, PO, AD and AMS received financial support of the Programa Institucional de Estrategias de Prevención en Obesidad y Diabetes del Instituto de Investigaciones Biomédicas-UNAM.

**Competing interests:** The authors have declared that no competing interests exist.

## Introduction

In mammals, 24 h fluctuations in numerous physiological variables, including metabolic parameters, ranging from circulating nutrient levels and substrate utilization to energy expenditure and thermogenesis, have been reported [1]. At the cellular level, the generation of circadian rhythmicity depends on a transcriptional-translational auto-regulatory feedback loop of the canonical core clock genes and their proteic products [2]. The clock genes are expressed abundantly and rhythmically in the circadian pacemaker located in the hypothalamic suprachiasmatic nucleus as well as in the peripheral oscillators that are present in nearly every tissue and organ [3]. The coupling between tissue-specific oscillators and the pacemaker is necessary for the control of circadian physiology.

Several studies have shown a reciprocal crosstalk between the circadian system and metabolism; critical genes associated with lipid and carbohydrate metabolism regulation, particularly the nuclear receptor superfamily, are expressed in a circadian manner in key metabolic tissues such as liver, muscle, brown adipose tissue (BAT), and white adipose tissue (WAT) in rodents [4, 5].

Experimental evidence indicates that the circadian molecular clock plays a fundamental role in energy homeostasis. The homozygous Clock mutant mice, in which the core *Clock* circadian gene is mutated, exhibit an attenuated rhythm food intake and develop hyperphagia, obesity, and different symptoms associated with metabolic syndrome, such as hyperglycemia, hyperlipidemia, and hepatic steatosis [6]. The deletion of essential components of the molecular clockwork leads to abnormal weight gain at a young age, as well as impairments in glucose metabolism and insulin hypersensitivity in mice [7, 8]. Conversely, the exposure to desynchronizing situations such as chronic jet-lag or long-term shifts occurring due to rotating shift works have adverse consequences on the circadian timing system and on various metabolic parameters. In murine models, changes in the light-dark cycle were found to produce an increase in weight gain, plasmatic leptin levels, and insulin/glucose ratio and to exacerbate cardiovascular diseases [9, 10]. In addition, chronic sleep disturbances alter glucose homeostasis; similar findings have been reported in humans [11]. These findings indicate that alterations in the regulation of circadian rhythmicity compromise homeostasis, leading to different metabolic diseases.

Conversely, metabolic alterations and obesity have an important impact on the generation and expression of circadian rhythmicity. Obese Zucker rats exhibit changes in the diurnal pattern of body temperature, locomotor activity, and feeding [12, 13]. In the obese (KK) and obese and diabetic (KK-A$^y$) mice models, the molecular clockwork of peripheral tissues (liver and visceral adipose tissue) is disrupted [14]. Numerous studies have indicated that the unrestricted access to high-fat diet has important effects on circadian behavior and the expression of neuropeptides in the hypothalamus and clock genes in the liver and adipose tissue in rodents [15]. The restriction of energy intake only to the sleep-phase produces changes in the phase relation between central and peripheral oscillators, resulting in increased adiposity, glucose intolerance, and dyslipidemia, a metabolic profile that exhibits close similarities to that observed in subjects with night-eating syndrome [16]. In contrast, in rodents, food consumption during the normal active phase prevents the disruption of the normal cellular metabolic program [17].

In western countries, the incidence of woman of reproductive age with obesity or overweight and having metabolic disorders has increased; in most of the cases, these conditions persist during gestation and perinatal development of offspring. Recent studies indicate that over 60% of pregnant women in México and the US exhibit obesity and metabolic alterations. Maternal metabolic state has a long-term effect on the development of offspring and is associated with increased risk of obesity and several other features associated with metabolic syndrome [18, 19].

The intrauterine environment is known to play a relevant role in influencing the susceptibility to certain chronic diseases in offspring. In response to the adverse environment, fetuses adapt their physiological development to maximize survival. These adaptations include resetting metabolic homeostasis set points, i.e., endocrine systems, and down-regulating growth, which is commonly manifested as an altered phenotype at birth [20, 21]. In addition, a major determinant of long-term disease risk is the degree of mismatch between pre- and post-natal environments [22]. Therefore, adaptive changes in fetal physiology that are relevant for survival *in utero* can be maladaptive in later life, contributing to disease risk throughout life when offspring are exposed to high-calorie food and low energy expenditure, among other factors [22].

Significant efforts have been undertaken to elucidate the mechanisms of fetal metabolic programming. Experimental evidence suggests that, at the peripheral level, maternal diet programs cause adipose tissue to promote hypertrophy, increase adipogenesis, and cause alterations in thermogenesis [23, 24], whereas, at the central level, the maternal condition affects the hypothalamic neural pathways that modulate appetite and satiety, as well as the circadian regulation of metabolic and molecular variables [25, 26].

The pathogenesis of human obesity and development of metabolic syndrome (MS) is not fully understood, in order to elucidate the mechanisms and develop new therapeutic strategies it is essential to have an appropriate animal model that share with human the most important aspects of the disease process. Rabbits is a widely used experimental model in biomedical research, has been proposed as an experimental alternative for the study of MS and its complications, such as atherosclerosis and coronary heart disease, which is the major cause of death in MS patients. Unlike rodents, rabbits have close similarities to humans cardiovascular and lipoprotein profile, with higher levels of apoB-containing low density lipoproteins, and abundant cholesteryl ester transfer protein in plasma, an important regulator of reverse cholesterol transport [27–29]. In addition, rabbits fed high fat and sugar diet develop many characteristics of MS observed in humans [30, 31].

On the other hand, rabbits are an ideal model for the study of transgenerational effect of maternal overnutrition in newborn metabolic regulation, due to the placental structure and materno-fetal blood flow interrelationships that is closer to the human, in comparison to other model, such as rodents. Humans have discoid and hemochorial type placenta, and the number of trophoblastic layers in the placental barrier as the border between fetal and maternal blood systems differs between species, in humans is hemomonochorial with only one layer, in rabbits is hemodichorial and in rodents is hemotrichorial [32, 33].

It has been documented in several animal models, that maternal overnutrition during pregnancy due to high fat and/or carbohydrates diet alters the metabolic future of the offspring, is possible that due to close relationship between the circadian system and metabolism, maternal overnutrition could alter key elements (oscillators, efectors or the coupling between them) of the system in charge of the generation and maintenance of circadian rhythmicity, which may occur prior to or simultaneously with the development of metabolic disorders. In order to address this issue, we used a rabbit model to determine whether maternal overnutrition leads to alterations in offspring development of diurnal rhythmicity of core body temperature and gross locomotor activity, as well as at homeostatic level on milk intake, growth, carbohydrates and lipids metabolism and molecules associated to thermogenesis, during lactation.

## Material and methods

The experiments were performed according to the National Institutes of Health Guide for the Care and Use of Laboratory Animals (NIH Pub. No. 86–23, revised 1996) and the Treatment of Animals in Research guidelines of the Instituto de Investigaciones Biomédicas, Universidad

Nacional Autónoma de México (UNAM). The protocol was reviewed and approved by the Animal Care and Use Committee of the Instituto de Investigaciones Biomédicas, UNAM, before the conduct of the study (ID: 198).

## Animals, diet, and experimental protocol

The study was conducted using the chinchilla strain of domestic rabbits (*Oryctolagus cuniculus*). Two groups of lactating rabbits were used, one obtained from does fed chow standard diet for rabbit (SD) and the second group obtained from does fed energy-rich high-fat and carbohydrate diet (HFCD). The animals were bred and maintained at the Instituto de Investigaciones Biomédicas, UNAM. Breeding female rabbits were housed in individual stainless-steel cages (120 × 60 × 45 cm) and maintained on a 16-h light/8-h dark cycle (the lights turned on at 09:00 h). The room temperature was maintained at 20 ± 2 ˚C with a relative humidity between 40% and 60%.

F1 was obtained from 24-week-old juvenile female rabbits (n = 20) fed chow SD for rabbit (providing 2542.6 Kcal/kg, 3.8% and 47.8% of kcal from fat and carbohydrates, respectively) (Conejo Ganador, Malta Cleyton, México) or with a HFCD (providing 2609.2 Kcal/kg, 5.6% and 52.6% of kcal from fat and carbohydrates, respectively). HFCD contained SD supplemented with 0.1% cholesterol (Sigma, USA), 4% soy oil (Sigma, USA), and 15% refined sugar (Great Value, México), providing 47% and 10% more energy from fat and carbohydrates than that obtained from SD. During the experimental time, food and water were available *ad libitum*.

At 20 weeks of age, does from both groups were mated with 24-month-old males of the same strain that were fed SD. SD females (n = 8) were fed SD during all experimental procedures. For HFCD females (n = 12), the supplemented diet was administered during eight consecutive weeks before mating. The conception rate of the HFCD does was improved by administering 30 UI of human chorionic gonadotropin (Chorulon, Intervet, México) 5 min before mating, whereas SD rabbits received only saline physiological solution. Spontaneous miscarriage has been associated with high fat intake [34] and also noted in our previous pilot experiments, this was avoided decreasing the frequency of the supplemented diet two weeks previous to mating to HFCD females, the HFCD was administrated once per week and the rest of the week does were fed with SD.

During pregnancy, does were housed individually. SD females continued with SD diet, whereas HFCD females were fed both diets on alternate days in order to avoid miscarriages as previously observed in pilot studies. Four days before the programmed date of parturition, an artificial burrow was placed in the cage of the pregnant rabbits. The burrow (28 × 29.5 × 30 cm high) was made of opaque polyvinyl chloride with a 14 cm diameter entrance. Sterile hay was placed in each maternal cage for building nests. Prior to the administration of the supplemented diet, SD and HFCD female rabbits exhibited similar body weight. Nevertheless, after 7 weeks of administration of supplemented diet, the HFCD does ingested significant more Kcal from fat and carbohydrates, as well as a significant increase in the body weight, and hyperglycemia and hyperlipidemia before mating and after the delivery, in comparison to SD females [35].

The day of parturition was defined as postnatal day (P) 0. The newborn rabbits were weighed at birth, marked on their ears for individual identification, and allowed to remain in the burrow with the mother for 6 h. Sixty-three rabbit pups obtained from 20 litters were divided into two groups: pups obtained from does fed SD, and pups obtained from over-nourished does fed HFCD. Each litter was adjusted to contain 6 newborn rabbits. Rabbits pups

were maintained under a 12:12 light–dark cycle (lights on at 12:00 h), room temperature, and relative humidity in the same ranges previously mentioned, during the experiment.

Thirty-two pups (SD, n = 15; HFCD, n = 17) were randomly assigned for the behavioral and physiological recording, with no more than two pups from the same litter. The newborn rabbits were transferred to the recording room, isolated from the rest of the colony, and placed individually in translucent polysulfide cages (48 × 27 × 20 cm) containing sterile nest material and corn cob bedding (Argo, México).

Newborn rabbits from both groups were nursed by foster lactating females fed SD. For maternal nursing, at the beginning of the light phase, pups were removed from their acrylic cages and placed inside the nest box; the sliding door that separates the cage in two compartments was opened to allow the foster mother access to the young to nurse. About 5 min after the does had finished nursing and had left the pups, the sliding door closed, and the pups were returned to their cages, until the same time next day. This procedure was repeated from P1 until P31. Gradual weaning was initiated from P32-35 (transition stage), thus in addition to nursing, in this stage, SD was also provided to all the rabbits *ad libitum*. The animals were weaned on P36, so they were only fed SD *ad libitum*.

## Milk intake and body weight

The daily amount of milk ingested by the pups was estimated by obtaining the body weights of all of the animals under study before and after nursing. Therefore, before the nursing, the urogenital area of the pups was gently rubbed to stimulate defecation and urination.

At the end of lactation, on P31, thirty-one individuals were randomly selected and killed at light onset (zeitgeber time (ZT) 00) and 12 h later (light offset, ZT12). No more than two pups from the same litter were included at one time point. Pups were removed from their cages, weighed, placed on a heating pad, and anesthetized by Sevoflurane inhalation. Blood samples were obtained by performing thoracotomy by cardiac puncture with a 25 G needle and 5 ml syringe that was inserted into the right ventricle, and 3 ml of blood was collected into test tubes sterile coated with silicone (BD Vacutainer 366668). Immediately, the brain, liver, stomach, kidney, heart, interscapular BAT, and mesenteric and retroperitoneal WAT were collected and weighed. WAT and BAT were stored at -70 ˚C until further analysis.

## Determination of serum Biochemical parameters

The blood samples were centrifuged at 3000 rpm for 15 min to obtain serum that was frozen at -70 ˚C for subsequent determination of the concentration of glucose, total cholesterol, low-density lipoproteins, high-density lipoproteins, free fatty acids, triglycerides, glycerol, urea, creatinine, albumin, total and conjugated bilirubin, aspartate aminotransferase, alanine aminotransferase, gamma glutamil transpeptidase, and creatinine phosphokinase. The serum samples were processed using spectrophotometric methods as previously described for rabbit pups [36] by using commercial enzymatic colorimetric assay kits (Randox Laboratories LTD., UK and Biosino Bio-technology & Science Inc.). The assays were performed as recommended by the manufacturers.

## Recordings and data analysis of gross locomotor activity and core body temperature

The locomotor activity and body temperature of individual rabbits were recorded simultaneously by using telemetry according to previously described methods [36–38]. On P8, the rabbit pups were anaesthetized by isoflurane inhalation (Inhalation anesthesia system; VetEquip Inc, USA; Sofloran Vet, Pisa Agropecuaria, México), and a transponder (G2 E-Mitter;

VitalView System, MiniMitter Respironics Inc., USA) was implanted *i.p.* under aseptic conditions. These sensors measured $15.5 \times 6.5$ mm and weighed 1.1 g, as previously reported [36–38]. After the pups were recovered from anesthesia, they were transferred to the recording room and placed in their cages equipped with a telemetry receiver (ER-4000 Energizer Receiver; MiniMitter Respironics Inc., USA). Four days after the surgical procedure, core body temperature and gross locomotor activity recordings were initiated. The data for both parameters were collected in 2-min bins by using a VitalView telemetry system (Respironics, MiniMitter Inc., USA). The parameters were recorded from P12 to P42.

## Determination of UCP1 and CIDEA gene expression

The uncoupling protein-1 (UCP1) and cell death-inducing DNA fragmentation factor-like effector A (CIDEA) mRNA levels in interscapular BAT and mesenteric fat (mWAT) and retroperitoneal fat (rWAT) were determined. Total RNA extraction was performed using TRIzol (Ambion by Life Technologies, CA, USA) according to manufacturer's protocols, followed by the determination of the concentration and purity. cDNA synthesis was performed using M-MLV reverse transcriptase enzyme and Oligo (dT) 12–18 primer (Invitrogen, Carlsbad, CA, USA).

The levels of mRNA were determined using real-time quantitative PCR by using 50 ng reverse-transcriptase product in 96-well optical plates by using an ABI Prism 7000 Sequence Detection System (Applied Biosystems, Foster City, CA, USA) and SYBR® Select Master Mix (Applied Biosystems, Life Technologies, CA, USA) with the following primers: UCP1; Fwd: 5□– TGCAGCCCTTATCCTCTTGC –3□, Rev: 5□– CTTGGATCTGTTGCCGGACT –3□; CIDEA Fwd: 5□– ATGGGCTCGAAGACAAAGCA –3□, Rev: 5□– CCAGAGCTTCCAGAGTCTTGT –3□; Hypoxanthine phosphoribosyltransferase (HPRT) Fwd: 5□– GCCCCAGCGTTGTGAT-TAGT –3□, Rev: 5□– CGAGCAAGCCTTTCAGTCCT –3□; and Cyclophilin A Fwd: 5□– CCCACCGTGTTCTTCGACAT –3□, Rev: 5□– ACCCTGGCACATAAACCCTG –3□.

PCR assays for each sample and analyzed gene were performed in duplicate under the following thermal cycling conditions: 50 ˚C for 2 min, 95 ˚C for 2 min, and then 40 cycles of 95 ˚C for 15 s and 60 ˚C for 1 min, followed by the default dissociation stage.

The relative quantification of UCP1 and CIDEA mRNA was based on the primer efficiency ($E = 10 (– 1/slope)$) and the threshold cycle values for each group and tissue sample, all compared with those from BAT, fed SD and killed at ZT00. Data are presented as the relative transcript levels compared with the HPRT and Cyclophilin A housekeeping genes, according to the equation for relative expression [39, 40].

## Data analysis

The time series obtained for core body temperature and gross locomotor activity were divided into three segments that were analyzed separately: *lactation* (P12 to 31); *transition* (P32 to 35) when, in addition to nursing, SD was also provided; and *weaning* (P36 to 42). To evaluate the rhythmicity in the rabbits' core body temperature and locomotor activity, we used a previously reported procedure [36, 37].

Rhythmicity in the rabbits' core body temperature was obtained following this procedure: (1) We minimize the effect of the developmental increase in temperature [41] adjusting the data to a straight line ($y = mx + b$) using the least-squares fitting method. The resulting straight line was subtracted from the original data and the increasing effect, represented by *m*, in reduced or eliminated. (2) We remove undesired noise using the Wave-shrink algorithm [42, 43]. We performed the Anscombe transformation [44] to translate the data distribution from Poisson to Gaussian, forcing the data noise to conform to a constant standard deviation; next,

we applied the wavelet transform with the Daubechies-8 wavelet [45] and apply minimal hard thresholding to the wavelet coefficients to remove those thresholds that concerned noise. Finally, we carried out the inverse wavelet transform to obtain a denoised data. (3) For energy analysis the Fast Fourier transform (FFT) was applied to the noise-free data, considering FFT can determine the amplitude, frequency and phase of every sinusoid present in the data [46]. The 20 components that contained the highest energy were selected for analysis. (4) To analyze the daily acrophase and nadir of the 24-h component we built a sequence of positive pulses of width $w$ that were spaced 24-h apart with the data maxima, as well as a sequence of negative pulses with the same characteristics as the data minima. The maxima and minima were obtained using the first and the second derivatives, respectively. The pulse sequences were positioned and shifted alongside all of the data to determine the acrophase and nadir, which corresponded to the positions at which $p$ was maximum, considering that $p$ is the absolute value of the sum of the products between the data and the pulse sequence. (5) Finally, the duration and amplitude of the anticipatory component were calculated. We defined anticipation as a constant increment of temperature over time. We employed the data corresponding to the first 5 hours of each 24-h segment to determine the positions at which the increment began and ended (duration) using the maximum and minimum values as well as the difference between both extreme values (amplitude). We defined four cases of analysis according to the position of the increment relative to the data mean: (a) it crossed the mean; (b) it was under the mean; (c) it was above the mean; and (d) it was not presented. Only cases (a) and (c) were considered anticipation.

The processing and analysis of locomotor activity was performed in the same way as for temperature, but the data adjustment was omitted because the activity pattern did not exhibit the same tendency as temperature during development. In addition, the noise was removed by Savitzky-Golay filtering [47] to preserve important details in the data, *i.e.*, the maxima and minima peaks.

The data are presented as the mean ± S.E.M for the daily locomotor activity, core body temperature, phases, and the duration and intensity of the anticipatory component; the differences associated with the maternal nutrition and/or age were tested using two-way ANOVAs for repeated measures, followed by the Scheffe post hoc test. In addition, we used linear regression in the cases of the daily average of activity and temperature.

For the weight of the organs and metabolic parameters, the values of the two groups were compared using two-way ANOVAs for the factors of group and time, followed by the Scheffe post hoc test (Statview, USA).

For UCP1 and CIDEA gene expression analysis, the differences between groups for each time (ZT 00 or 12) were tested using one-tailed unpaired *t*-test (with Welch's correction when variance between groups was statistically significant) or with one-tailed Mann-Whitney test. One-way ANOVA followed by Fisher's LSD multiple comparison test or Kruskal-Wallis followed by Dunn's multiple comparison test was used to analyze differences among groups with regard to the time of the day (ZT 06; GraphPad Prism, USA).

In all cases, $p < 0.05$ was considered statistically significant, and differences are indicated in each figure.

## Results

### Body weight and milk intake

The 2-way ANOVA revealed a significant effect among groups and age on the body weight of newborn rabbits during lactation (Group: $F_{1, 1932} = 4.5$; $p \leq 0.03$; Age: $F_{35, 1932} = 375.4$; $p \leq 0.0001$; Interaction: $F_{35, 1932} = 2.4$; $p \leq 0.0001$). At birth (P0), the animals of the HFCD

group were significantly lighter (21%) than the SD pups (Fig 1A and 1B). The body weight of both the groups increased gradually during lactation, exhibiting similar weights from P2 to P30 (Fig 1C). At the end of lactation (P31) and during the transition stage, the pups of the HFCD group exhibited a significant increase in weight compared with that of the SD pups, reaching a final weight on P35 of 608.2 ± 24 g and 539.5 ± 33.8 g, respectively (Fig 1C).

With respect to milk intake, the 2-way ANOVA revealed a significant effect among groups and age (Group: $F_{1, 1743} = 41.7$; $p \leq 0.0001$; Age: $F_{34, 1743} = 41.7$; $p \leq 0.0001$; Interaction: $F_{34, 1743} = 3.1$; $p \leq 0.0001$). The main effects were observed at the end of lactation and during the transition stage, where the HFCD pups ingested 188% more maternal milk than the SD pups (Fig 1D).

## Organs weight and Biochemical and metabolic characterization

At the end of lactation, we did not find any difference between the weight of the organs in the groups at any time of the day (S1 Table). Both groups exhibited no difference on the plasma concentrations of analytes related to glucose and lipid metabolism at the end of lactation (Table 1). In addition, the kidney and heart damage markers did not differ between the groups (Table 2). However, the liver damage markers, GGT and total bilirubin, exhibited a significant increase in the HFCD group compared to that in the SD group (Table 2).

## Locomotor activity rhythm

At the end of lactation and during transition and weaning, the HFCD pups had higher levels of locomotor activity, of about 5%-12%, than that of the pups in the SD group (Fig 2A). Significant effects associated with maternal nutrition on the average daily locomotor activity of newborn rabbits were observed during lactation (Group: $F_{1, 599} = 4.5$; $p = 0.03$; Age: $F_{19, 599} = 1.5$; $p = NS$; Interaction: $F_{19, 599} = 0.4$; $p = NS$). At the beginning of the lactation stage (P12), the mean activity of SD and HFCD pups was similar between both groups (13.4 ± 0.6 mov/2 min and 13.7 ± 0.7 mov/2 min, respectively). At the end of lactation (P31), the average activity remained at the same level in SD pups (13.7 ± 0.9 mov/2 min), whereas it increased significantly in the HFCD pups (15.4 ± 0.8 mov/2 min) compared to that in SD rabbits at the same stage (Fig 2B). In the transition stage, the animals continued to exhibit significant changes associated with maternal nutrition (Group: $F_{1, 28} = 12.5$; $p = 0.001$; Age: $F_{3, 28} = 4.4$; $p = 0.01$; Interaction: $F_{3, 28} = 1.4$; $p = NS$); the mean activity of SD pups was 13.4 ± 0.7 mov/2 min, whereas that of the HFCD pups significantly increased (16.3 ± 0.7 mov/2 min; Fig 2B). During weaning, the animals continued to exhibit significant changes associated with maternal nutrition (Group: $F_{1, 56} = 4.5$; $p = 0.03$; Age: $F_{7, 56} = 0.9$; $p = NS$; Interaction: $F_{7, 56} = 0.7$; $p = NS$); the mean activity of the SD group was 13.5 ± 0.5 mov/2 min, whereas that of the HFCD group was 15.1 ± 0.5 mov/2 min (Fig 2B).

With respect to the temporal pattern of locomotor activity, during lactation, the rabbits in the SD group exhibited a characteristic diurnal pattern (Fig 3A), in which the activity started to rise above the 24-h mean, approximately 1 h before the scheduled time of access to the lactating doe for nursing. Following this episode, the average activity dropped below the 24-h mean and remained at this low level. A second rise in activity was observed close to the transition to the dark phase of the cycle. In contrast, the rabbits in the HFCD group exhibited an atypical temporal pattern of gross locomotor activity during lactation and transition stages, which differed considerably from that of the SD group (Fig 3A and 3D). All the pups in the HFCD group exhibited activity above the 24-h activity mean approximately 3 h before the scheduled time of access to the female for nursing. After this episode, the average body activity

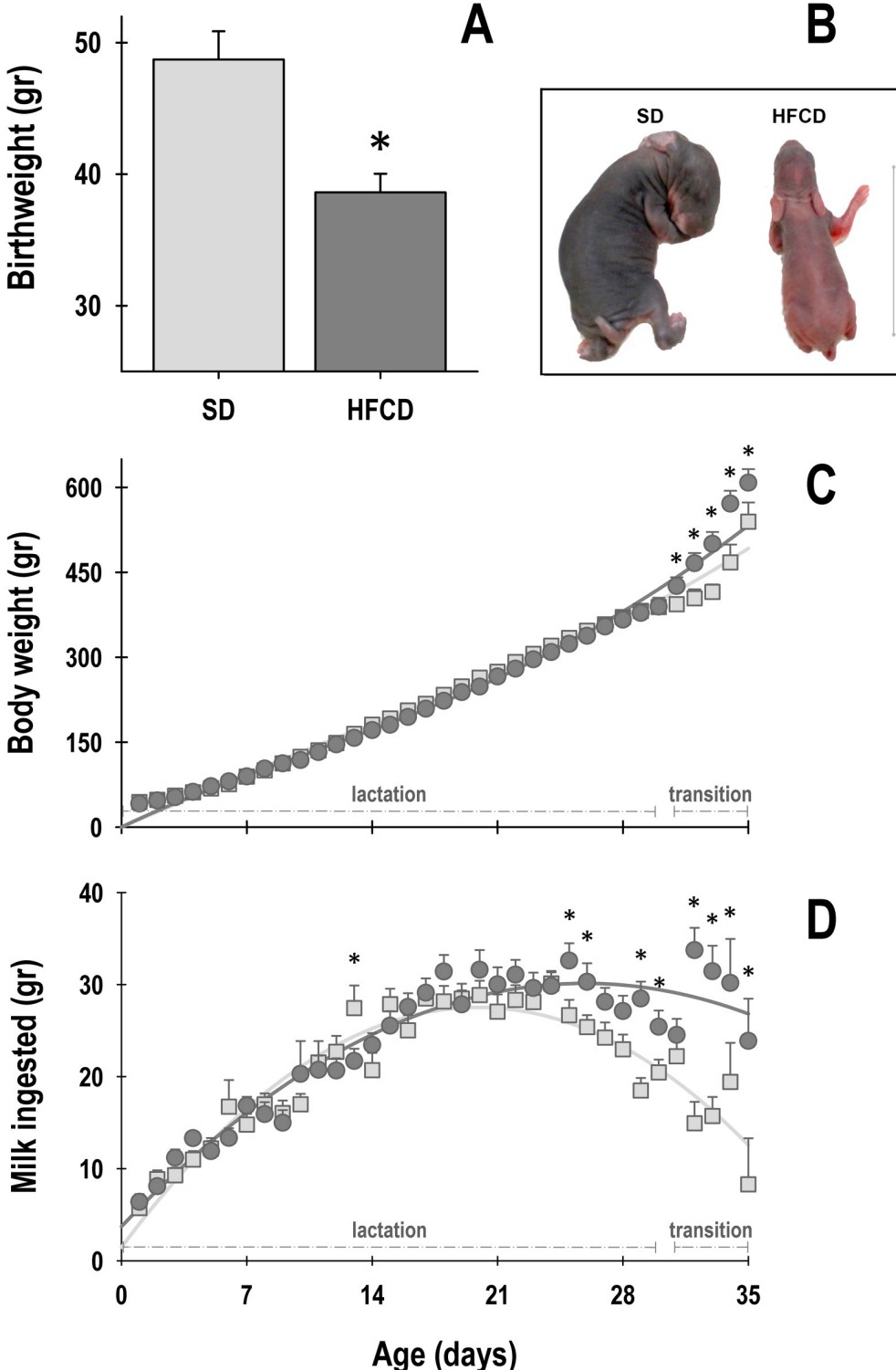

**Fig 1. Body weight and milk intake.** (A) Mean body weight at birth from rabbits obtained from females fed a standard diet (SD, light gray bars) and females fed high-fat and carbohydrate diet (HFCD, dark gray bars). (B) Photographs of representative rabbits at birth obtained from SD and HFCD mothers. Daily average of body weight (C) and milk ingestion (D) of SD (light gray squares) and HFCD (dark gray circles) rabbits. From postnatal days 12 to 31, the pups were nursed every 24 h at ZT 00; from postnatal days 32 to 35, they were nursed as well as fed standard diet (transition stage). Mean ± SEM. * indicates a significant difference ($p < 0.05$) *vs*. SD.

**Table 1. Daily average of serum glucose and lipids levels.**

|  | light phase | | dark phase | | p values | | |
|---|---|---|---|---|---|---|---|
|  | SD | HFCD | SD | HFCD | Group | ZT | Group x ZT |
| *Glucose metabolism* | | | | | | | |
| Glu | 5.3±0.2 | 5.3±0.1 | 5.7±0.1 | 5.4±0.2 | 0.3 | 0.2 | 0.3 |
| *Lipid metabolism* | | | | | | | |
| Chol | 11.3±0.1 | 10.8±0.8 | 12.6±1 | 13.3±1.5 | 0.9 | 0.9 | 0.6 |
| LDL | 3.5±0.4 | 3.2±0.2 | 3.9±0.2 | 3.8±0.3 | 0.6 | 0.1 | 0.7 |
| HDL | 1.8±0.2 | 1.7±0.1 | 1.7±0.1 | 1.9±0.3 | 0.8 | 0.7 | 0.3 |
| VLDL | 29.2±7.7 | 29.7±2.4 | 34.5±6.9 | 45.9±9.8 | 0.4 | 0.1 | 0.4 |
| FFA | 14.7±3.8 | 15.7±3.5 | 13.3±1.9 | 14.5±2.3 | 0.7 | 0.7 | 0.9 |
| TG | 1.5±0.4 | 1.3±0.1 | 1.2±0.2 | 1.3±0.2 | 0.7 | 0.5 | 0.5 |
| GLY[#] | 190.6±17.2 | 244.1±27.8 | 189.3±16.8 | 284±80.4 | 0.1 | 0.7 | 0.7 |

Plasmatic levels of different analytes associated to glucose and lipids metabolism, in rabbit pups obtained from females fed standard diet (SD) or high fat and carbohydrate diet (HFCD), at the end of the lactancy. Results of one-way ANOVA obtained for the differences associated with maternal condition (group) and the time (zeitgeber time, ZT). Mean ± SEM.

[#]μmol/L

dropped below the 24-h mean and remained at this low level for only 6 h. Bouts of activity were evident during the remaining light phase and the entire dark phase (Fig 3D).

At weaning, the temporal pattern of activity in both the groups was reorganized. In the SD group, the first rise associated with nursing exhibited an important decrement (Fig 3A),

**Table 2. Kidney, heart and liver damage markers.**

|  | light phase | | dark phase | | p values | | |
|---|---|---|---|---|---|---|---|
|  | SD | HFCD | SD | HFCD | Group | ZT | Group x ZT |
| *Kidney damage markers* | | | | | | | |
| Urea[#] | 14.7±14.9 | 1.3±3.7 | 11.6±1.5 | 10.3±1.3 | 0.9 | 0.2 | 0.8 |
| Creatinine[†] | 68.6±2.8 | 77.4±6.2 | 63.7±5.8 | 76.8±7 | 0.1 | 0.7 | 0.7 |
| *Heart damage marker* | | | | | | | |
| CPK§ | 442.2±36.7 | 701.7±62.5 | 718.2±186.9 | 795.4±57.9 | 0.1 | 0.1 | 0.4 |
| *Liver damage markers* | | | | | | | |
| GGT§ | 2±0.3 | 3±0.3 | 2.6±0.2 | 2.8±0.2 | **0.02**[*] | 0.5 | 0.09 |
| Total Bil[†] | 4.2±0.6 | 6.4±0.5 | 4.5±0.6 | 6±0.7 | **0.02**[*] | 0.5 | 0.9 |
| Con Bil[†] | 0.7±0.02 | 0.8±0.07 | 0.9±0.1 | 0.9±0.1 | 0.4 | 0.2 | 0.7 |
| AST§ | 13.7±3.2 | 17.9±2.7 | 11.8±1.7 | 14.4±1.1 | 0.2 | 0.3 | 0.7 |
| ALT§ | 19.8±2.4 | 22.6±3.2 | 22.3±1.1 | 22.4±2.6 | 0.6 | 0.7 | 0.6 |
| AST/ALT§ | 0.6±0.1 | 0.8±0.08 | 0.7±0.1 | 0.7±0.08 | 0.4 | 0.3 | 0.6 |
| Albumin‡ | 42.2±2.9 | 41.9±1.5 | 39±1.1 | 38.9±2.9 | 0.9 | 0.2 | 0.9 |

Plasmatic levels of different analytes associated to kidney, heart and liver damage markers, in newborn rabbit obtained from females fed standard diet (SD) or high fat and carbohydrate diet (HFCD). Results of one-way ANOVA obtained for differences associated with maternal condition (group) and the time (zeitgeber time, ZT). Mean ± SEM.

[*] indicates a significant difference ($p < 0.05$) *vs*. SD.

[#]mmol/L,

[†]μmol/L,

[‡]g/L, and

§U/L.

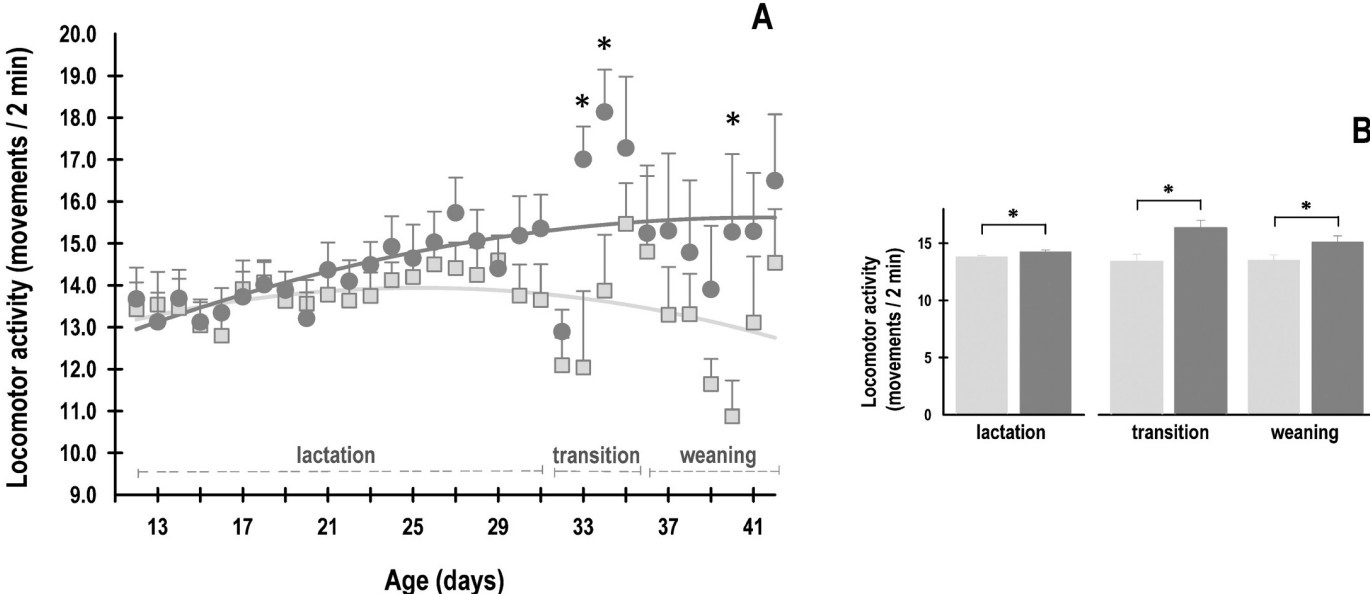

**Fig 2. Daily average of gross locomotor activity.** (A) The mean activity from rabbits obtained from females fed a standard diet (light gray squares) or high-fat and carbohydrate diet (dark gray circles). From postnatal days 12 to 31, the pups were nursed every 24 h at ZT 00; from postnatal days 32 to 35, they were nursed as well as fed standard diet (transition stage); and from postnatal day 36, the rabbits were weaned, and all the animals received standard diet ad libitum (weaning). (B) The mean values of activity in each stage. Mean ± SEM. * indicates a significant difference ($p < 0.05$) *vs*. SD.

whereas both the bouts of activity exhibited an important increment in their duration in the HFCD group (Fig 3D). Fourier analysis revealed that, during transition and weaning stages, the 12-h component was present in 100% of SD animals and the 24-h component was present in the 5 most energetic components in 86% of the cases (Fig 3B); in contrast, in the HFCD animals, the 12-h component was present in 100% of the cases, but the 24-h component was present in the 5 most energetic components only in 72% of the cases (Fig 3E).

The time when the locomotor activity remained above the daily average level was similar between the groups and also exhibited significant changes associated with age (Group: $F_{1, 203}$ = 0.003; $p$ = NS; Age: $F_{30, 203}$ = 3.2; $p \leq 0.0001$; Interaction: $F_{30, 203}$ = 1.4; $p$ = NS). During lactation and transition stages, the locomotor activity of the animals of the SD group remained above the average during 10 h 42 min ± 10 min nd below the average during 13 h 17 min ± 10 min (Fig 3C). For the animals of the HFCD group, the time when the activity remained above the average was 10 h 48 min ± 17 min, and the time expended below the average was 13 h 10 min ± 28 min (Fig 3F).

With regard to the time when the maximal activity was noted during lactation, ANOVA did not reveal significant effects (Group: $F_{1, 599}$ = 1.3; $p$ = NS; Age: $F_{19, 599}$ = 1.5; $p$ = NS; Interaction: $F_{19, 599}$ = 0.9; $p$ = NS). The acrophase of the diurnal rhythm of activity in the rabbits in the SD and HFCD groups during lactation occurred at 03:07 h ± 22 min and 03:41 h ± 21 min, respectively. During the transition stage (Group: $F_{1, 28}$ = 0.09; $p$ = NS; Age: $F_{3, 28}$ = 0.6; $p$ = NS; Interaction: $F_{3, 28}$ = 0.8; $p$ = NS) and weaning (Group: $F_{1, 56}$ = 0–09; $p$ = NS; Age: $F_{7, 56}$ = 0.9; $p$ = NS; Interaction: $F_{7, 56}$ = 0.7; $p$ = NS), the acrophases of the activity pattern of both groups were similar. The nadir of the activity rhythm was different according to maternal nutrition during lactation (Group: $F_{1, 599}$ = 4.1; $p$ = 0.04; Age: $F_{19, 599}$ = 0.8; $p$ = NS; Interaction: $F_{19, 599}$ = 1.7; $p$ = 0.03), in the transition stage (Group: $F_{1, 28}$ = 1.5; $p$ = NS; Age: $F_{3, 28}$ = 4.5; $p$ = 0.01; Interaction: $F_{3, 28}$ = 3; $p$ = NS), and in weaning stage (Group: $F_{1, 56}$ = 1.2; $p$ = NS; Age: $F_{7, 56}$ =

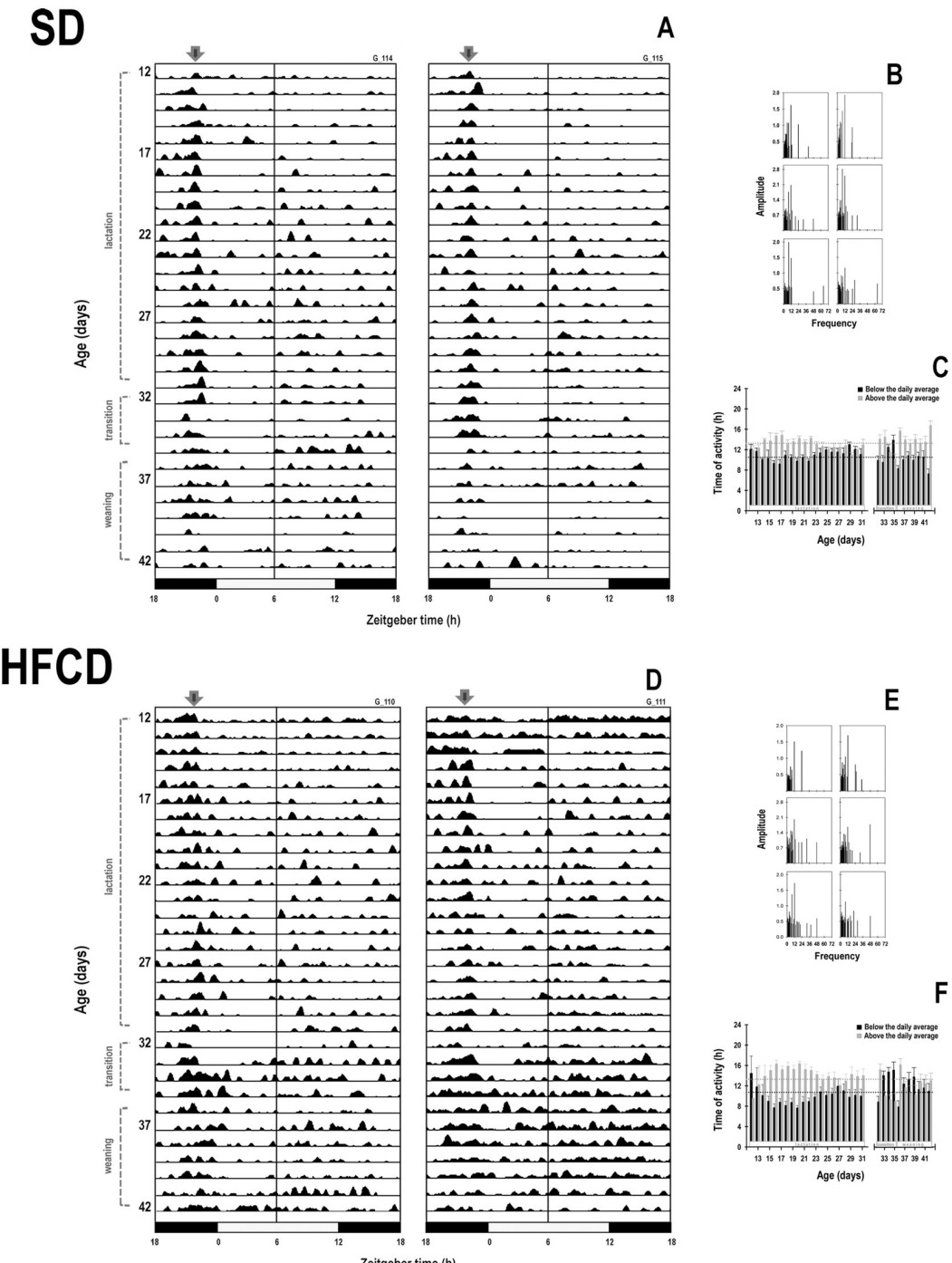

**Fig 3. Temporal patterns of gross locomotor activity.** Representative daily profiles of activity rhythm as measured using biotelemetry in newborn rabbits obtained from females fed standard diet (A) or high-fat and carbohydrate diet (D). From postnatal days 12 to 31, the pups were nursed every 24 h at ZT 00 (indicated by the black arrow); from postnatal days 32 to 35, they were nursed as well as fed with standard diet (transition stage); and from postnatal day 36, the rabbits were weaned, and all the animals received standard diet ad libitum (weaning). On the right panel, representative graphics of the Fourier spectra obtained from both groups of animals at different stages of development, in newborn rabbits obtained from females fed a standard diet (B) or high-fat and carbohydrate diet (E). On bottom right panel the time for which the activity remained above (gray bars) or below (black bars) the average daily level, in newborn rabbits obtained from females fed with a standard diet (SD, C) or with a high-fat and carbohydrate diet (HFCD, F). Horizontal dotted lines represent the mean time durations of high (gray) and low (black) activity.

1.1; $p$ = NS; Interaction: $F_{7, 56}$ = 1.8; $p$ = NS). During lactation, the minimal activity occurred at 07:41 h ± 24 min and 06:39 h ± 20 min in the SD and HFCD groups, respectively. During the transition stage, the acrophase and nadir in both groups showed a similar tendency. At weaning, these parameters became irregular in both groups.

The duration of the pups' anticipatory activity of nursing did not reveal significant changes (Group: $F_{1, 594}$ = 3.5; $p$ = NS; Age: $F_{19, 594}$ = 1.5; $p$ = NS; Interaction: $F_{19, 594}$ = 1.1; $p$ = NS). During lactation, the SD and HFCD groups exhibited a rise in the anticipatory activity for 128 ± 03 min and 120 ± 03 min, respectively (Fig 4, top left panel). In the transition stage, both groups exhibited a similar anticipatory rise in temperature (Group: $F_{1, 28}$ = 0.2; $p$ = NS; Age: $F_{3, 28}$ = 0.2; $p$ = NS; Interaction: $F_{3, 28}$ = 1.9; $p$ = NS) for 126 ± 11 min and 117 ± 14 min for the SD and HFCD groups, respectively (Fig 4, middle left panel). Notably, during weaning, the duration of the anticipatory activity decreased (Group: $F_{1, 55}$ = 1.8; $p$ = NS; Age: $F_{7, 55}$ = 0.9; $p$ = NS; Interaction: $F_{7, 55}$ = 0.3; $p$ = NS). The SD group showed an anticipation duration of 101 ± 08 min, whereas the anticipation in the HFCD pups lasted for 85 ± 08 min (Fig 4, bottom left panel).

The intensity of the anticipatory rise in locomotor activity was not significantly different (Group: $F_{1, 594}$ = 0.04; $p$ = NS; Age: $F_{19, 594}$ = 1.2; $p$ = NS; Interaction: $F_{19, 594}$ = 0.9; $p$ = NS). During lactation, the magnitude of the anticipatory rise in the SD group was 28.9 ± 1.1 mov/2 min, and the HFCD group exhibited a similar magnitude of anticipatory rise at 28.9 ± 1.2 mov/2 min (Fig 4, top right panel). In the transition stage, both groups exhibited a similar anticipatory rise in locomotor activity (Group: $F_{1, 28}$ = 0.7; $p$ = NS; Age: $F_{3, 28}$ = 0.9; $p$ = NS; Interaction: $F_{3, 28}$ = 0.8; $p$ = NS): the SD group showed a rise of 23 ± 2.2 mov/2 min and the HFCD group showed a rise of 57.5 ± 35.9 mov/2 min (Fig 4, middle right panel). During weaning, the intensity of the anticipatory rise was similar in both groups (Group: $F_{1, 55}$ = 0.2; $p$ = NS; Age: $F_{7, 55}$ = 1.3; $p$ = NS; Interaction: $F_{7, 55}$ = 0.5; $p$ = NS): the SD group showed an increase of 17.3 ± 1.1 mov/2 min, and the HFCD group showed an increase of 18.7 ± 3.9 mov/2 min (Fig 4, bottom right panel).

## Core body temperature

Significant effects associated with maternal nutrition and age on the average daily temperature of newborn rabbits were observed during lactation (Group: $F_{1, 599}$ = 15; $p$ = 0.0001; Age: $F_{19, 599}$ = 69.5; $p$ = < 0.0001; Interaction: $F_{19, 599}$ = 0.5; $p$ = NS). The average daily core body temperature of the rabbit pups of both the groups increased significantly with age (Fig 5A). During lactation, the mean temperature was 38.9 ± 0.01 ˚C for SD pups and 39 ± 0.01 ˚C for HFCD pups. In the transition stage, the animals continued to exhibit significant changes associated with maternal nutrition (Group: $F_{1, 28}$ = 8.3; $p$ = 0.005; Age: $F_{3, 28}$ = 4.9; $p$ = 0.005; Interaction: $F_{3, 28}$ = 0.4; $p$ = NS). During this stage, the animals exhibited a conspicuous increase in temperature reaching 39.4 ± 0.07 ˚C in the SD group and 39.7 ± 0.09 ˚C in the HFCD group (Fig 5B). During weaning, the animals again continued to exhibit significant changes associated with maternal nutrition (Group: $F_{1, 56}$ = 9.1; $p$ = 0.003; Age: $F_{7, 56}$ = 0.3; $p$ = NS; Interaction: $F_{7, 56}$ = 0.2; $p$ = NS), in which the temperature of the SD animals was 39.7 ± 0.01 ˚C, whereas that of the HFCD animals was 40 ± 0.06 ˚C (Fig 5B). At the end of lactation and during transition and weaning, the HFCD pups had an increase in core body temperature by approximately 0.3 ˚C at the end of the experiment, compared with that in the SD group (Fig 5A).

With respect to the temporal pattern of the core body temperature during lactation, the rabbits in the SD group exhibited a characteristic diurnal pattern (Fig 6A), in which the core body temperature started to rise above the 24-h mean approximately 2 h before the scheduled time of access to the lactating doe for nursing. Following this episode, the average body temperature dropped below the 24-h mean and remained at this low level for about 3 h. A second rise in

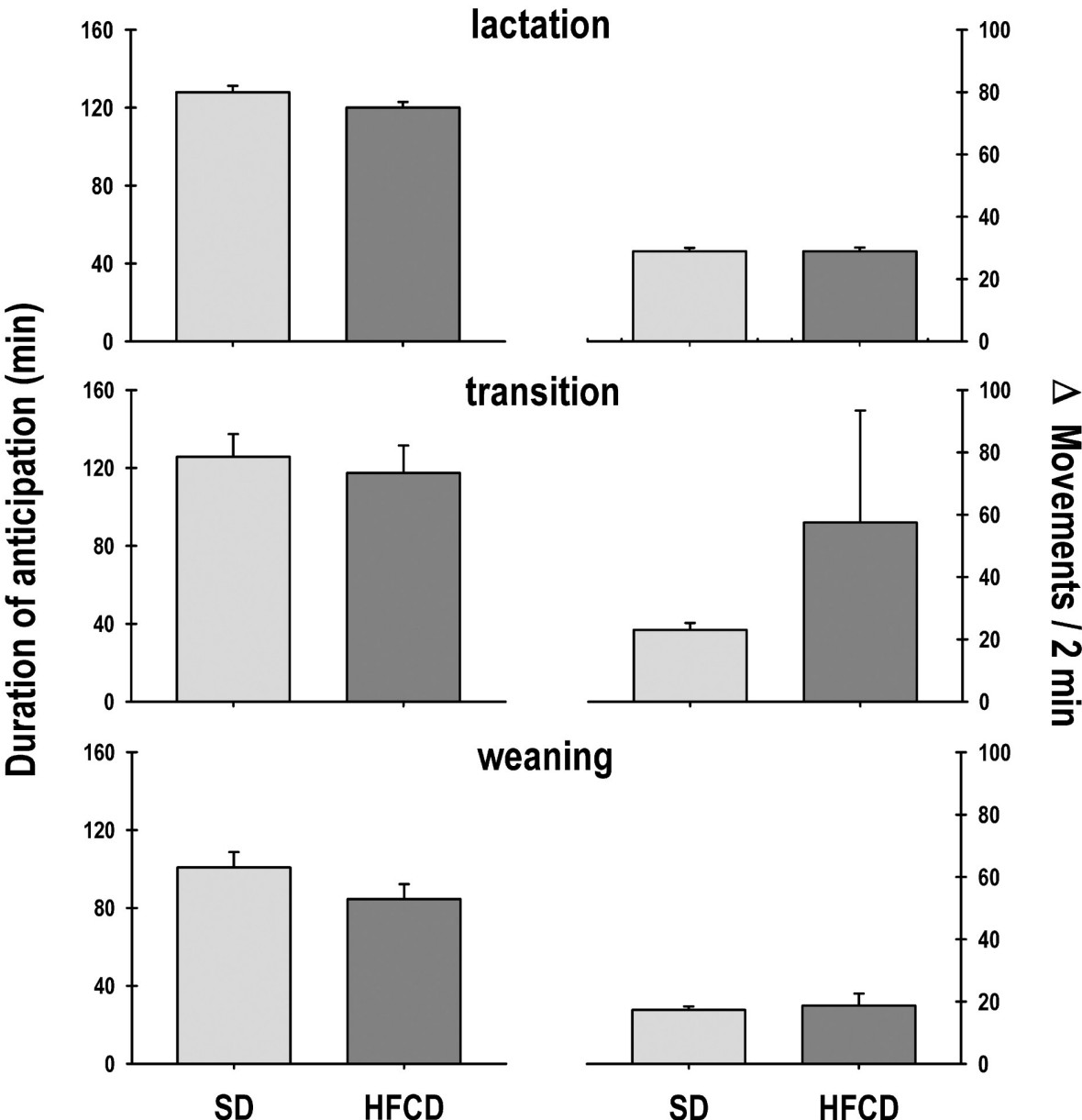

**Fig 4. The anticipatory component of gross locomotor activity.** Graphics of the duration (left panel) and intensity (right panel) of the anticipation in locomotor activity of newborn rabbits obtained from females fed standard diet (SD) or high-fat and carbohydrate diet (HFCD). From postnatal days 12 to 31, the pups were nursed every 24 h at ZT 00; from postnatal days 32 to 35, they were nursed as well as fed standard diet (transition stage); and from postnatal day 36, the rabbits were weaned, and all the animals received standard diet ad libitum (weaning). Mean ± SEM.

temperature was observed approximately 2 h after the transition to the dark phase of the cycle, and the temperature remained above the average level during 6 h. In transition and weaning stages, the temporal pattern of temperature in the SD animals was reorganized, in which the first rise associated with nursing exhibited an important decrement (Fig 6A). During lactation, Fourier analysis revealed that, for the SD animals, the 12-h component was present in 94% of the cases and the 24-h component was present in the 5 most energetic components in 64% of

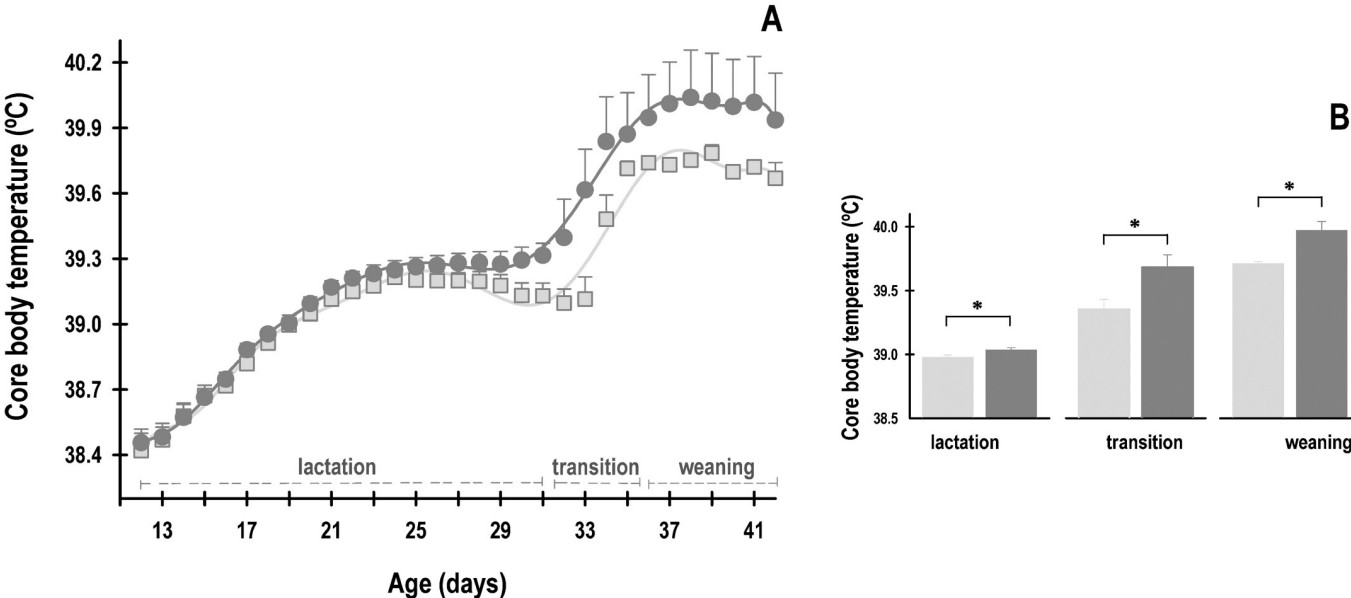

**Fig 5. Daily average of core body temperatures.** (A) The mean temperature from rabbits obtained from females fed standard diet (light gray squares) or high-fat and carbohydrate diet (dark gray circles). From postnatal days 12 to 31, the pups were nursed every 24 h at ZT 00; from postnatal days 32 to 35, they were nursed as well as fed standard diet (transition stage); and from postnatal day 36, the rabbits were weaned, and all the animals received standard diet ad libitum (weaning). (B) The mean values of temperature at each stage. Mean ± SEM.

the cases (Fig 6B). In contrast, the rabbits in the HFCD group exhibited an atypical temporal pattern of core body temperature during lactation and transition stages, which differed considerably from those of the SD group (Fig 6D). All the pups in the HFCD group exhibited temperatures above the 24-h mean temperature approximately 5 h before the scheduled time of access to the female. After this episode, the average body temperature dropped below the 24-h mean and remained at this low level only for 1.5 h. A second rise in temperature was observed approximately 3.5 h after the transition to the dark phase of the cycle, and the temperature remained above the average level until the next nursing episode (Fig 6D). In the transition stage and during weaning, the temporal pattern of temperature in the HFCD animals was also reorganized: the first and second rises of temperature exhibited an important increment in their duration (Fig 6D). During lactation, Fourier analysis revealed that, for the HFCD animals, 12-h component was present in 94% of the cases and the 24-h component was present in the 5 most energetic components in 61% of the cases (Fig 6E).

The ANOVA revealed significant effects associated with maternal nutrition, age, and the interaction between these factors on the time for which the temperature remained above the average daily level (Group: $F_{1, 619} = 73.2$; $p \leq 0.0001$; Age: $F_{31, 619} = 20.2$; $p \leq 0.0001$; Interaction: $F_{31, 619} = 1.9$; $p = 0.001$). During lactation and transition stages, the temperature of the animals in the SD group remained above the average for 6 h ± 15 min and below the average for 17.9 h ± 18 min (Fig 6C). In contrast, the animals of the HFCD group exhibited a significant increase in the time for which the temperature remained above the average (12.1 h ± 19 min) and below the average (11.9 h ± 19 min) (Fig 6F).

With regard to the time when the maximal temperature was noted during lactation, ANOVA revealed significant effects associated with maternal nutrition and age (Group: $F_{1, 599} = 10.9$; $p = 0.0001$; Age: $F_{19, 599} = 4.5$; $p \leq 0.0001$; Interaction: $F_{19, 599} = 0.9$; $p = NS$). The acrophase of the diurnal rhythm of the temperature in the rabbits of the SD and HFCD groups

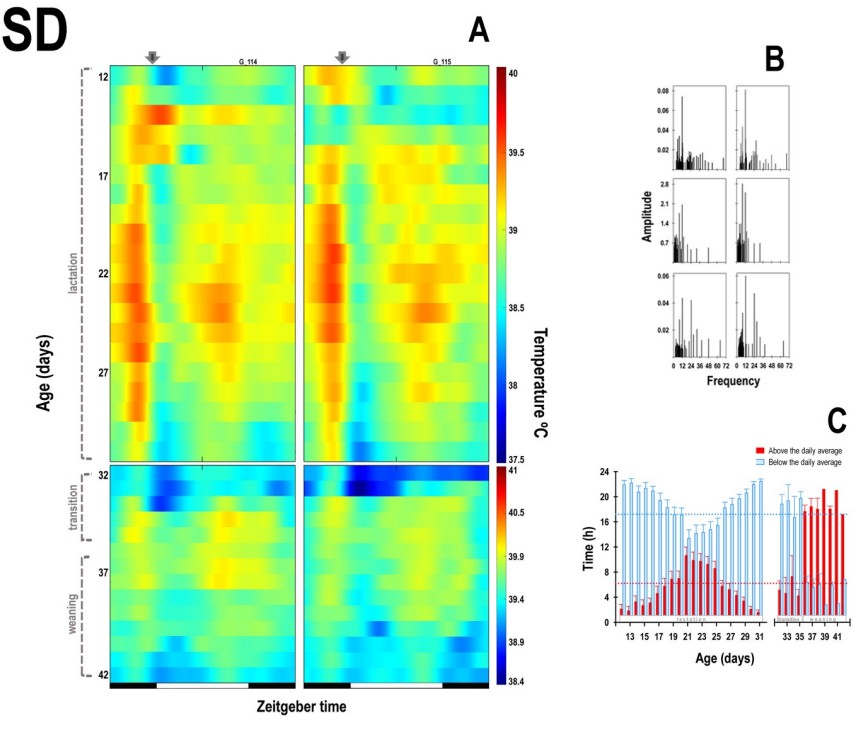

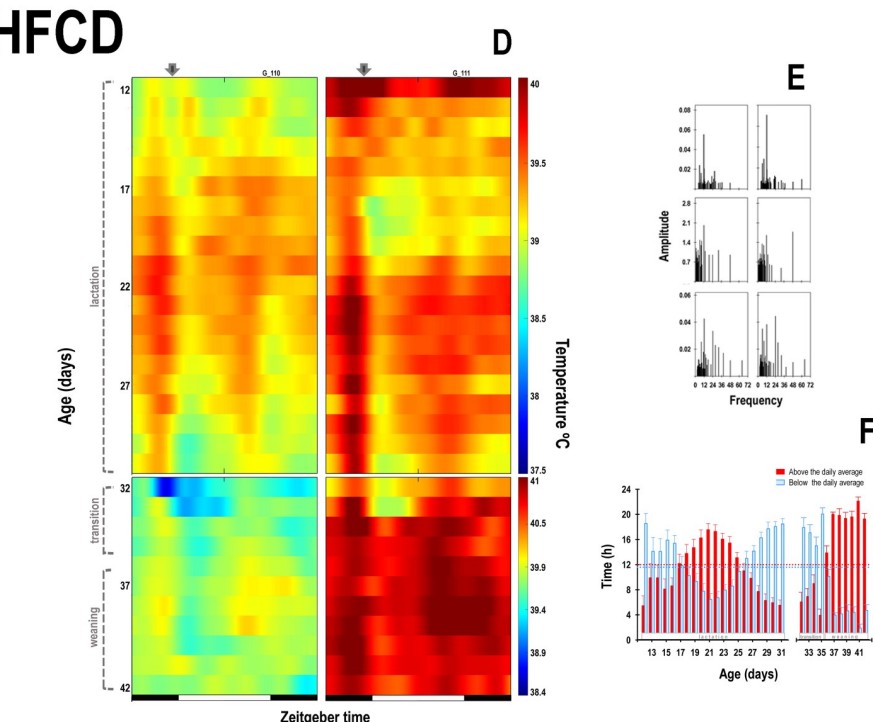

**Fig 6. Temporal patterns of core body temperature.** Representative daily profiles of temperature rhythm as measured using biotelemetry, in newborn rabbits obtained from females fed a standard diet (A) or high-fat and carbohydrate diet (D). From postnatal days 12 to 31, the pups were nursed every 24 h at ZT 00 (indicated by the black arrow); from postnatal days 32 to 35, they were nursed as well as fed a standard diet (transition stage); and from postnatal day 36, the rabbits were weaned, and all the animals received a standard diet ad libitum (weaning). The right panel contains representative graphics of the Fourier spectra obtained from both groups of animals at the different

stages of development, in newborn rabbits obtained from females fed a standard diet (B) or a high-fat and carbohydrate diet (E). On bottom right panel the time for which the temperature remained above (red bars) or below (blue bars) the average daily level, in newborn rabbits obtained from females fed with a standard diet (SD, C) or with a high-fat and carbohydrate diet (HFCD, F). Horizontal dotted lines represent the mean time durations of high (red) and low (blue) activity.

during lactation occurred at 0:15 h ± 12 min and 01:20 h ± 15 min, respectively. Therefore, the animals of the HFCD group exhibited a delay of 1.1 h in the acrophase compared to the SD pups (Fig 7A and 7B). During the transition (Group: $F_{1, 28} = 0.1$; $p$ = NS; Age: $F_{3, 28} = 4.2$; $p = 0.01$; Interaction: $F_{3, 28} = 0.2$; $p$ = NS) and weaning (Group: $F_{1, 56} = 1.2$; $p$ = NS; Age: $F_{7, 56} = 2.2$; $p = 0.04$; Interaction: $F_{7, 56} = 1.8$; $p$ = NS) stages, the achrophases of both groups were similar (Fig 7A and 7B). Nevertheless, on P36, the new phase relation of the maximal temperature with the onset of the dark phase was evident in the SD rabbits; in contrast, the HFCD rabbits did not reach a stable phase relation at the end of the experiment (Fig 7A and 7B). The nadir of the rhythm was similar in both groups during the lactation (Group: $F_{1, 599} = 0.08$; $p$ = NS; Age: $F_{19, 599} = 1.5$; $p$ = NS; Interaction: $F_{19, 599} = 0.7$; $p$ = NS), transition stage (Group: $F_{1, 28} = 1.5$; $p$ = NS; Age: $F_{3, 28} = 4.5$; $p = 0.01$; Interaction: $F_{3, 28} = 3$; $p$ = NS), and weaning (Group: $F_{1, 56} = 1.2$; $p$ = NS; Age: $F_{7, 56} = 1.1$; $p$ = NS; Interaction: $F_{7, 56} = 1.8$; $p$ = NS) stages.

The duration of the pups' anticipation of nursing revealed significant changes associated with maternal nutrition and age (Group: $F_{1, 594} = 4.2$; $p = 0.04$; Age: $F_{19, 594} = 4.6$; $p \leq 0.0001$; Interaction: $F_{19, 594} = 1.4$; $p$ = NS). During lactation, the SD and HFCD groups exhibited an anticipatory rise in temperature during 131 ± 04 min and 120 ± 03 min, respectively. Therefore, the pups of the HDFC group exhibited a significant decrease in the anticipatory rise in temperature compared with those of the SD groups (Fig 8, top left panel). In the transition stage, both groups exhibited a similar anticipatory rise in temperature (Group: $F_{1, 28} = 1.2$; $p$ = NS; Age: $F_{3, 28} = 3.2$; $p = 0.04$; Interaction: $F_{3, 28} = 0.9$; $p$ = NS) during 136 ± 15 min and 116 ± 13 min for the SD and HFCD groups, respectively (Fig 8, middle left panel). Notably, during weaning, the duration of the anticipatory rise decreased in both groups (Group: $F_{1, 55} = 1.3$; $p$ = NS; Age: $F_{7, 55} = 4.2$; $p = 0.001$; Interaction: $F_{7, 55} = 0.9$; $p$ = NS). The SD group had a duration of 100 ± 9 min, and the anticipation in the HFCD pups lasted 115 ± 10 min (Fig 8, bottom left panel).

The intensity of the anticipatory rise in core body temperature revealed significant changes associated with maternal nutrition, age, and the interaction between both factors (Group: $F_{1, 594} = 20.8$; $p \leq 0.0001$; Age: $F_{19, 594} = 5.8$; $p \leq 0.0001$; Interaction: $F_{19, 594} = 1.6$; $p = 0.05$). During lactation, the magnitude of the anticipatory rise in the SD group was of 0.26 ± 0.009 ˚C, whereas the pups in the HFCD group exhibited a significant decrement in the magnitude of the anticipatory rise (0.2 ± 0.008 ˚C) compared with that of the remaining groups (Fig 8, top right panel). In the transition stage, both groups exhibited a similar anticipatory rise in temperature (Group: $F_{1, 28} = 1.1$; $p$ = NS; Age: $F_{3, 28} = 1.6$; $p$ = NS; Interaction: $F_{3, 28} = 1.2$; $p$ = NS): the SD group showed a rise of 0.39 ± 0.05 ˚C, and the HFCD group showed a rise of 0.32 ± 0.05 ˚C (Fig 8, middle right panel). During weaning, the intensity of the anticipatory rise in temperature decreased in both groups (Group: $F_{1, 55} = 0.2$; $p$ = NS; Age: $F_{7, 55} = 1.3$; $p$ = NS; Interaction: $F_{7, 55} = 0.5$; $p$ = NS): the SD showed an increase of 0.17 ± 0.01 ˚C, and the HFCD group showed an increment of 0.17 ± 0.01 ˚C (Fig 8, bottom right panel).

## UCP1 and CIDEA levels

In order to determine if the alteration in core body temperature was related to an increase in thermogenesis, the gene expression of biomarkers was evaluated in adipose tissue. In the three

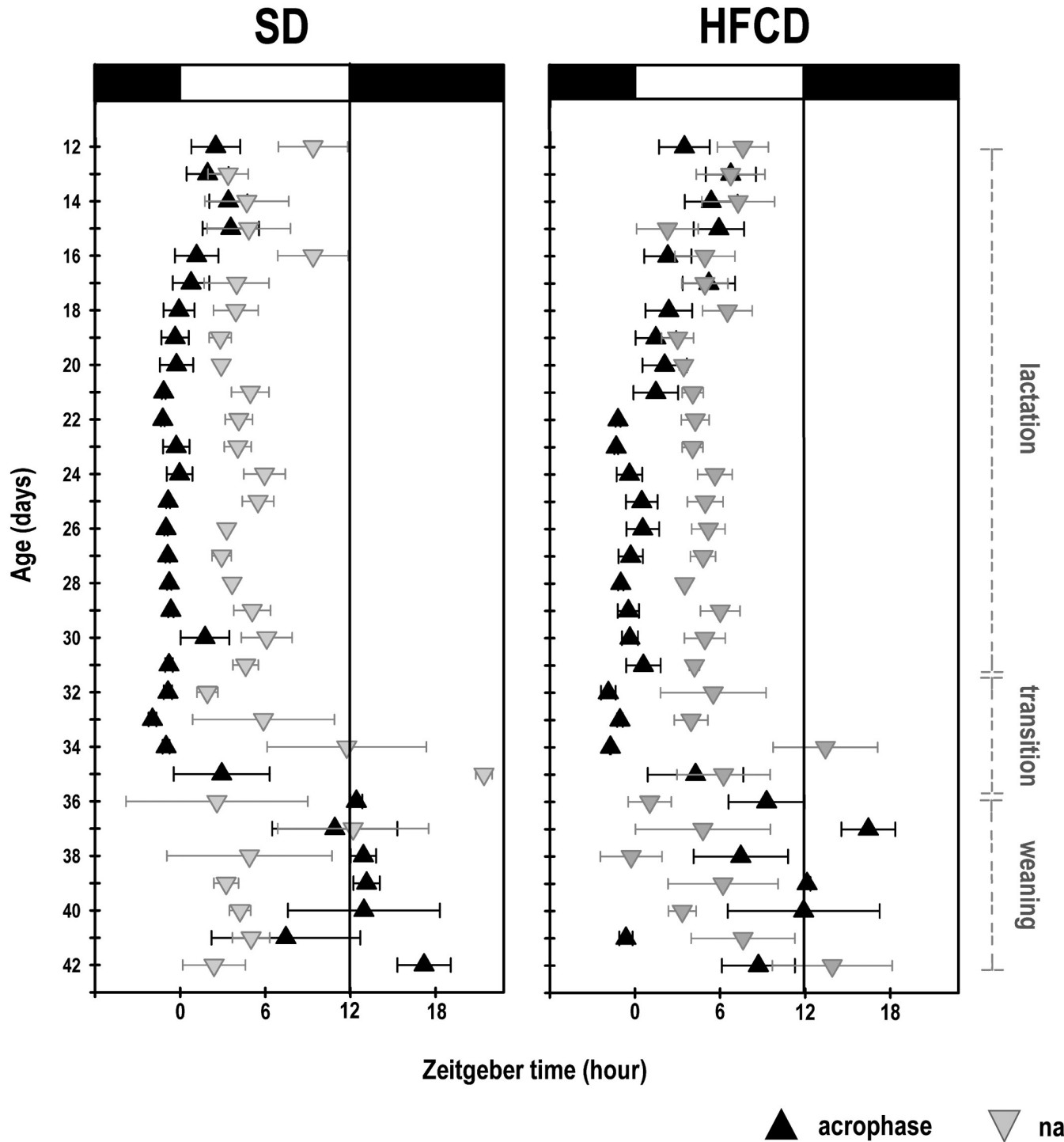

**Fig 7. Daily phases of core body temperature.** The daily acrophases (black triangles) and nadirs (gray triangles) of the diurnal pattern of core body temperature in newborn rabbits obtained from females fed standard diet (A) or high-fat and carbohydrate diet (B). From postnatal days 12 to 31, the pups were nursed every 24 h at ZT 00 (indicated by the gray line); from postnatal days 32 to 35, they were nursed as well as fed with standard diet (transition stage); and from postnatal day 36, the rabbits were weaned, and all the animals received standard diet ad libitum (weaning). The black line indicates one half of the cycle. Mean ± SEM.

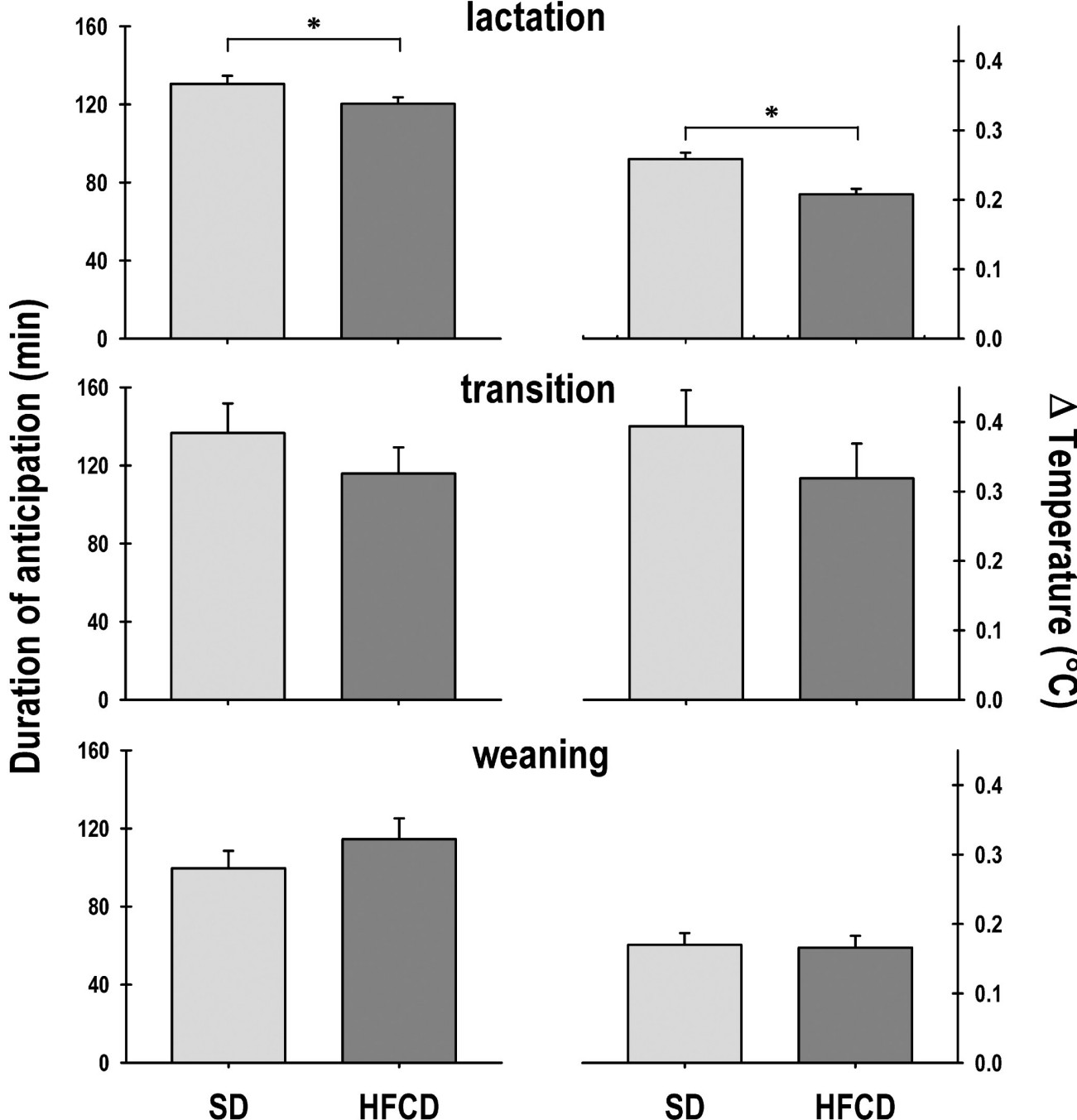

**Fig 8. The anticipatory component of core body temperature.** Graphics showing the duration (left panel) and intensity (right panel) of the anticipation in core body temperature of newborn rabbits obtained from females fed a standard diet (SD) or a high-fat and carbohydrate diet (HFCD). From postnatal days 12 to 31, the pups were nursed every 24 h at ZT 00; from postnatal days 32 to 35, they were nursed as well as fed the standard diet (transition stage); and from postnatal day 36, the rabbits were weaned, and all the animals received the standard diet ad libitum (weaning). Mean ± SEM. * indicates a significant difference ($p < 0.05$) *vs*. SD.

adipose tissue pads, UCP1 gene expression was greater in the light phase than in the dark phase. In particular, it was significantly different for the HFCD group; in contrast, for the SD group, the difference between time was statistically significant only in mWAT (Fig 9, left panel). In contrast, CIDEA mRNA expression was significantly higher in the dark phase in both groups (Fig 9, right panel).

UCP1 and CIDEA mRNA levels in BAT (Fig 9) were, respectively, 2.5- and 0.5-fold greater in the HFCD group than in the SD group at both times, indicating that pups of the HFCD group had more thermogenically active brown adipocytes.

With regard to the process of browning in WAT, as expected, UCP1 and CIDEA mRNA levels were, respectively, expressed between 12 and 900 times more in BAT than in WAT depending on the time of the day, except for CIDEA in rWAT that was expressed 0.2 to 2 more times than in BAT (Fig 9). Nevertheless, in mWAT, UCP-1 at ZT 00 and CIDEA at both times (Fig 9) were expressed significantly less in the HFCD group than in the SD group, suggesting less beige adipocytes or browning.

In rWAT, no differences in UCP1 and CIDEA gene expression was observed between the SD and HFCD groups, in neither of the two times of the day that were evaluated (Fig 9).

## Discussion

These results indicate that the European rabbit is a useful model for studying the effect of fetal environment and the regulation of circadian physiology.

Most literature reports indicate that maternal nutrition has a direct impact on the birth weight of the offspring; nevertheless, the effects differ considerably: studies in rodents showed that the intake of high-fat diet during pregnancy in some cases reduced offspring birth weight [48–50], in some cases increased birth weight [51], and in some, had no effect [52]. In the rabbit model, maternal chronic intake of HFCD had a significant effect on the body weight of lactating rabbits during early postnatal development, since HFCD offspring exhibited reduced body weight at birth, followed by an accelerated growth, such that, at the end of lactation, the pups were significantly heavier than the SD pups; similar findings have been reported in rats [48, 53]. In different models and in humans, adverse intrauterine factors have been shown to cause low birth weight and faster growth during postnatal development, leading to an increase in the susceptibility to obesity and the development of pathologies related to metabolic syndrome in later life [54–56].

Maternal over-nutrition seems to alter the central appetite regulators, as early as in lactation, since HFCD rabbits exhibited marked hyperphagia at the end of this stage. Similar findings have been reported in murine models, in which maternal hypocaloric restriction or the intake of high-fat diet altered the ingestion behavior of the offspring; this has been associated with the changes in hypothalamic pathways [57, 58]. Under normal conditions, the adipose-derived hormone leptin and insulin directly regulate the hypothalamic levels of neuropeptide Y (NPY), agouti-related protein (AgRP), pro-opiomelacortin (POMC), and cocaine and amphetamine-related transcripts (CART) to inhibit food intake and increase energy expenditure via the long form of leptin receptor. The offspring obtained from malnourished dams exhibited down-regulated anorexigenic peptides such as POMC and CART in early postnatal stages, whereas the mRNA of orexigenic peptides NPY and AgRP are up-regulated, favoring a higher energy intake [57]. Further studies are needed to determine if hyperphagia observed in rabbit pups is related to the changes in this food intake signaling pathway due to maternal over-nutrition.

Despite the increase of maternal milk ingestion in the HFCD pups, the animals did not exhibit alterations associated to carbohydrate and lipid metabolism at the end of lactation.

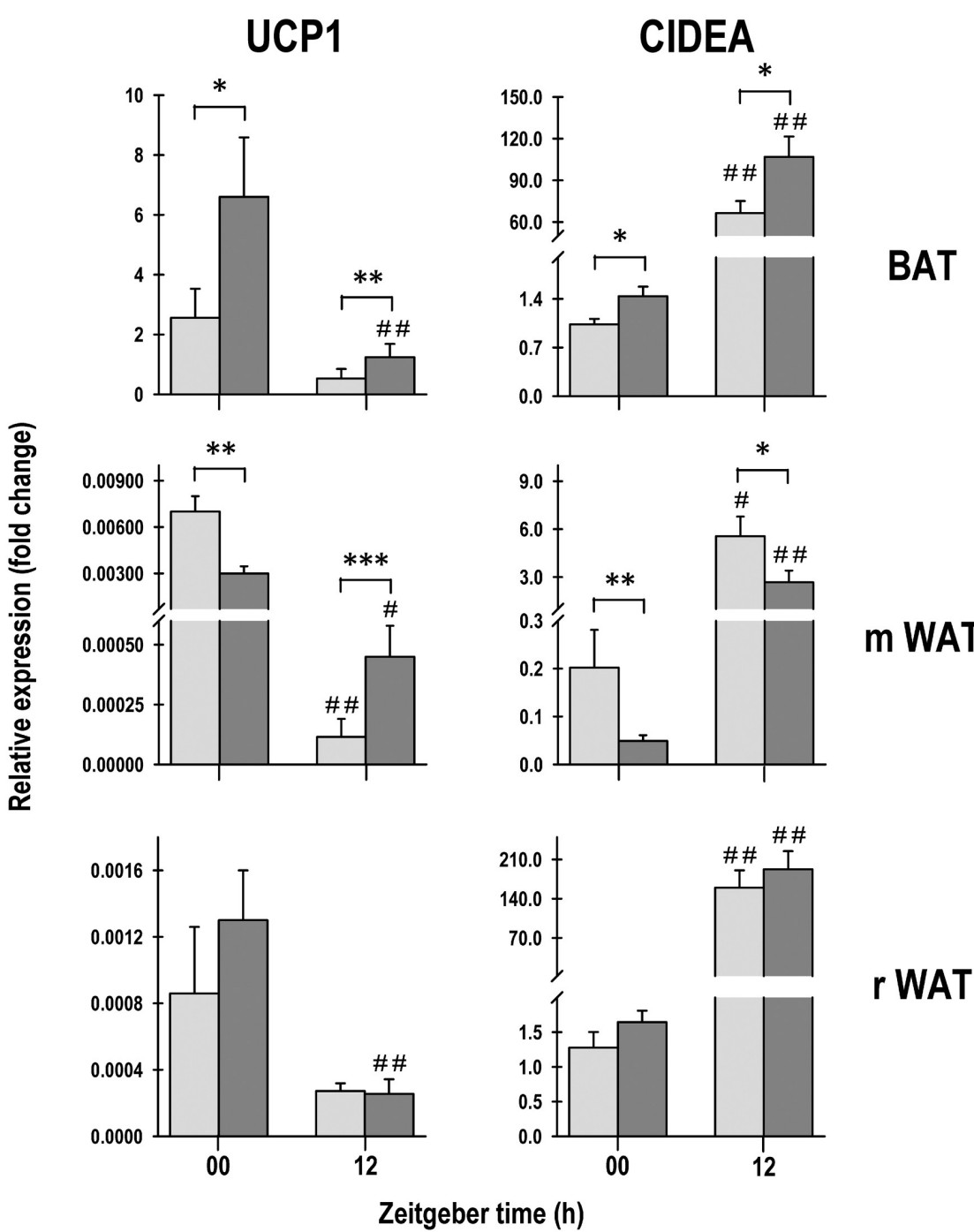

**Fig 9. UCP1 and CIDEA mRNA levels.** Relative abundance of UCP1 in BAT, mesenteric WAT, and retroperitoneal WAT, and of CIDEA in BAT mesenteric WAT and retroperitoneal WAT of rabbit pups fed standard diet (SD, gray bars) and high-fat and carbohydrate diet (HFCD, black bars) evaluated at light onset (ZT 00) and 12 h later at light offset (ZT 12). Data are presented as mean ± SEM. Statistically significant differences are indicated between SD and HFCD pup rabbits for each time condition, * $p < 0.05$ and **$p < 0.001$ and between light onset and offset for each experimental group (SD and HFCD), # $p < 0.05$ and # # $p < 0.001$.

However, determining the long-term effects of maternal nutrient intake on the metabolic profile of offspring is important, because animals could present the first signs and symptoms associated with the development of metabolic syndrome in adulthood.

In relation to the diurnal expression of circadian rhythmicity in offspring, at the behavioral level, newborn rabbits obtained from HFCD mothers exhibited clear hyperactivity and changes in the temporal pattern of gross locomotor activity, with a marked decrement of the 24-h frequency and changes in the architecture of the diurnal rhythm characterized by a significant increment of activity during the light phase, the phase of the cycle that corresponds to the resting phase of this nocturnal species. Similar findings have been reported in rodents, in which maternal malnutrition, such as the intake of low-protein diets during pregnancy, increased the level of activity during the resting phase [58, 59].

The most conspicuous effects were observed in the average core body temperature and its temporal regulation. To our knowledge, this is the first report on the multigenerational effect of maternal nutrition and changes in temperature regulation in offspring, since rabbit pups obtained from over-nourished mothers exhibited a significant increase in the daily average core body temperature, during the three stages under study.

It is well known that the *Oryctolagus cuniculus* has an unusual pattern of maternal care, the female give birth in burrow and only visit their young briefly for a few minutes, once approximately every 24 h to nurse [60, 61]. The young anticipate this vital event with increased arousal, uncovering from the nest material and an increase in body temperature one to two hours before the mother's arrival [37, 62]. This is important in enabling the young to prepare for the highly competitive daily scramble for nipples, which they typically locate in seconds and without direct behavioral assistance from their mother [63]. In core body temperature was evident a significant decrement in the duration and intensity of the circadian-regulated anticipatory component in HFCD pups, this effect was evident only during lactation, developmental stage in which the 24-h regulation has a crucial role in the survival of offspring that reside in subterranean burrows under constant dark conditions [61], with eyelids closed and when the visual system is not yet functional [64], indicating that the circadian time keeping system may be compromised by maternal nutritional status.

Literature reports show contradictory results about the differences on temperature between lean and obese subjects; some reports indicate lower average wrist temperature in obese women [65] and obese Zucker rats [66], and others reveal higher average core body temperature and fingernail-bed in obese humans [67, 68] and core temperature in dogs [69]. The increase in body temperature might be associated with a higher resting metabolic heat production owing to fat-free mass that accompanies excessive adiposity in obese subjects. The increase in adipose tissue provides an insulating barrier to conductive heat flow and reduces the ability to respond to changes in the core temperature [68].

Core body temperature in endothermic mammals is maintained through muscle-shivering thermogenesis or basal metabolism with the participation of BAT, through the high amounts of mitochondria brown adipocytes have, where UCP1 is expressed to uncouple respiration from energy production, resulting in the generation of heat [70]. Other thermoregulatory protein highly expressed in brown adipocytes includes CIDEA, which modulates UCP1 [71] and promotes the enlargement of lipid droplets by enabling one droplet to transfer its contents to another droplet [72]. Interscapular brown adipocytes of rabbit pups of the HFCD group underwent severe thermogenesis that could be reflected in the augmented core body temperature of these animals, suggesting an adaptive mechanism leading to the low birth weight. The distribution of interscapular BAT in rabbits exhibits close similarities to those reported in other species [73, 74].

The expression of UCP1 and CIDEA in WAT is also used as browning markers, since the emergence of beige adipocytes in WAT, adipocytes containing several small fat droplets but also a great mitochondrial density [71], represents an adaptation process to increase thermogenic demand, reducing adverse effects of WAT and improving metabolic health [70, 75] so browning induction has even been developed as treatment for obesity-related metabolic diseases. The gene expression of these browning markers in WAT, UCP1 and CIDEA, was lower in the HFCD group than in the SD group (except for UCP1 at ZT 12) in mWAT. This could suggest that white mesenteric adipocytes of the HFCD group have a lesser ability to become beige (browning process). Thus, rabbit pups of the HFCD group could have a greater risk for developing diseases related to the metabolism of carbohydrates and lipids in adult life. In fact, some authors have also noted that HFD feeding decreases UCP-1 expression in subcutaneous WAT, suggesting that beige adipocytes are displaced by white adipocytes to store fat in WAT under excess energy intake [76].

Other mechanisms are known to increase UCP1 expression/activation in BAT. Cold exposure increases the activity of nerves of the afferent sympathetic nervous system that innervate BAT, with the release of noradrenaline, in turn and finally inducing thermogenesis through the induction of UCP1 [77]. The induction of UCP1 can also be achieved by factors such as a high amount of dietary fat, particularly polyunsaturated fatty acids, salmon protein hydrolysate via bile acid induction, an excess of high energy intake, functional food components (capsicinoids, ginger, curcumin, menthol, epigallocatechin gallate, resveratrol, cocoa flavonoids, among others), sympathomimetics compounds [caffeine, ephedrine, etc) and some pharmacological agents [77]. Regular exercise training may also induces thermogenic activity through UCP1 expression in interscapular BAT in cases where animals consume a high-fat diet, or when exercise is combined with cold exposure [78].

In addition to the significant increase of body weight during the lactation stage, the rabbit pups of the HFCD group also exhibited significant changes in the temporal profile of their core temperature, mainly in the architecture and the quality of the 24-h rhythm. Experimental data suggest that the robustness of the circadian rhythm of core body temperature is negatively correlated with body weight, since obese humans, dogs, and mice had less robust rhythms compared to their counterparts with normal weight [65, 69, 79, 80].

The homeostatic and chronostatic functions of body temperature are is regulated by the hypothalamic preoptic area and SCN, respectively [81]. Experimental evidence indicates that the maternal nutritional condition alters the cytoarchitecture of these hypothalamic structures in the offspring [82, 83] and causes long-lasting disruption in the diurnal expression of clock genes in the liver and heart of the offspring [48]. Further studies are needed to determine whether the exposure to suboptimal environment, such as HFCD, during intrauterine development in rabbits has an impact on the hypothalamic regulation of temperature, on the molecular clock-work of the central pacemaker, and on the peripheral oscillators.

These results clearly indicate that maternal over-nutrition alters homeostatic and chronostatic regulation in the offspring at physiological and behavioral levels. Further studies are needed to determine accurately whether alterations in the expression of rhythmicity are associated with changes in the organization of the circadian system.

## Supporting information

**S1 Table. Organs weights.**
(PDF)

## Acknowledgments

We thank Monserrat Sordo, Luz M. Chiu and Pamela E. Vega for their excellent technical assistance.

## Author Contributions

**Conceptualization:** Erika Navarrete, Rodrigo Montúfar-Chaveznava, Patricia Ostrosky-Wegman, Ivette Caldelas.

**Data curation:** Rodrigo Montúfar-Chaveznava, Ivette Caldelas.

**Formal analysis:** Erika Navarrete, Andrea Díaz-Villaseñor, Rodrigo Montúfar-Chaveznava, Ivette Caldelas.

**Funding acquisition:** Rodrigo Montúfar-Chaveznava, Ivette Caldelas.

**Investigation:** Erika Navarrete, Andrea Díaz-Villaseñor, Georgina Díaz, Ana María Salazar, Patricia Ostrosky-Wegman, Ivette Caldelas.

**Methodology:** Andrea Díaz-Villaseñor, Georgina Díaz, Ana María Salazar, Ivette Caldelas.

**Project administration:** Ivette Caldelas.

**Resources:** Rodrigo Montúfar-Chaveznava.

**Software:** Rodrigo Montúfar-Chaveznava.

**Supervision:** Ivette Caldelas.

**Validation:** Erika Navarrete, Rodrigo Montúfar-Chaveznava, Ivette Caldelas.

**Visualization:** Erika Navarrete, Ivette Caldelas.

**Writing – original draft:** Erika Navarrete, Rodrigo Montúfar-Chaveznava, Patricia Ostrosky-Wegman, Ivette Caldelas.

**Writing – review & editing:** Andrea Díaz-Villaseñor, Georgina Díaz, Ana María Salazar, Rodrigo Montúfar-Chaveznava, Patricia Ostrosky-Wegman, Ivette Caldelas.

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
