## [Decision Letter · Decision Letter 0]

10 Jan 2020

PONE-D-19-28260

Misadjustment of diurnal expression of core temperature and locomotor activity in lactating rabbits associated with maternal over-nutrition before and during pregnancy

PLOS ONE

Dear Dr. Caldelas,

Thank you for submitting your manuscript to PLOS ONE. After careful consideration, we feel that it has merit but does not fully meet PLOS ONE’s publication criteria as it currently stands. Therefore, we invite you to submit a revised version of the manuscript that addresses the points raised during the review process.

Please read the reviews below and respond accordingly. In particular, however,  there should be less emphasis on the circadian aspect of the paper, as it appears that many aspects of the clock and clock function are not the focus of the reported investigation, even though it does fit with previous studies focusing on the circadian regulation. Additionally, there should be more detail on the statistical analysis.. The manuscript also has some grammatical errors, which should be remedied.

We would appreciate receiving your revised manuscript by Feb 24 2020 11:59PM. To enhance the reproducibility of your results, we recommend that if applicable you deposit your laboratory protocols in protocols.io, where a protocol can be assigned its own identifier (DOI) such that it can be cited independently in the future. For instructions see: http://journals.plos.org/plosone/s/submission-guidelines#loc-laboratory-protocols

We look forward to receiving your revised manuscript.

Kind regards,

Paul A. Bartell

Academic Editor

PLOS ONE

Journal Requirements:

Reviewers' comments:

Reviewer's Responses to Questions

**Comments to the Author**

1. Is the manuscript technically sound, and do the data support the conclusions?

Reviewer #1: Yes

Reviewer #2: No

2. Has the statistical analysis been performed appropriately and rigorously? 

Reviewer #1: Yes

Reviewer #2: No

3. Have the authors made all data underlying the findings in their manuscript fully available?

Reviewer #1: Yes

Reviewer #2: Yes

4. Is the manuscript presented in an intelligible fashion and written in standard English?

Reviewer #1: Yes

Reviewer #2: Yes

5. Review Comments to the Author

Reviewer #1: The manuscript investigates a novel question using a unique experimental model. The experiment appears to have been well controlled and made key observations of behavior and body temperature rhythms. The next obvious step is investigation of the molecular clock. The authors clearly state this and do not over-interpret the data.

Resolution of some of the figures is very poor and impossible to read some of the smaller figures.

I am not sure Table 1 needs to be reported or could be a supplement, but I will leave that to the editor and author to decide.

741: Access

Reviewer #2: General Comments:

This paper measures daily patterns of locomotor activity and body temperature, along with body temperature. The paper appears to be a companion paper to another submitted paper that measured body weight and fertility, based on the citation in the methods. While the paper focuses on an interesting area of fetal programming, I have several concerns about the experimental methods and presentation of results. Specifically, the paper focuses heavily on the circadian aspect of this experiment, but there is really no examination of any circadian rhythms or circadian clock gene expression. Additionally, there is no description of how acrophase, the only response related to daily rhythms, was determined. Overall, the statistical analysis was poorly performed, and P values of most responses were not presented. Several clarifications must be made, and an overhaul of the statistics and presentation of results must be made before I can consider accepting this paper for publication. Additionally, of the results presented only a small number of them are actually related to the objective of studying the effects of maternal nutrition on diurnal rhythms, and most are related to metabolism (UCP1 and CIDEA expression and day vs. night concentration of blood metabolites). The title and introduction, and objectives of the paper should more accurately reflect the responses measured.

Line-by-line comments:

Abstract:

43- Please more clearly describe the ‘anticipatory component” of body temperature or use different terminology. Same for “energetic components of the rhythm”. The meaning of these phrases is non-obvious

Introduction:

117 – Replace with “Women of reproductive age”

120- Be more specific about what conditions these women exhibit

123- Within the introduction there must be justification for why rabbits are used as a model species for maternal overnutrition, and specifically why they can be a good model for humans, since research in human subjects is directly described in the introduction

134 – There is a lot of information in the introduction describing the effects of maternal overnutrition on the circadian clock. However, there is no measurement of the circadian clock gene expression or circadian rhythms of energy metabolism within this paper. There should be a better connection between how the research presented in this paper relates to the circadian clock, or this information should be removed and the introduction should focus more on behavioral and body temperature outputs

143- A hypothesis must be provided

144 – Objective does not include anything about UCP1 or CIDEA expression levels or blood metabolites, so why were these measured? Please add objectives related to these outcomes. Also, circadian rhythms in behavioral and physiologic markers weren’t measured. In order to characterize a true circadian rhythm, responses must be measured in consistent lighting (i.e. 24 dim light). Please refer to the rhythms measured (really only locomoter activity) as a diurnal or daily rhythm.

156 – Must state the experimental design in the methods

164 – Please provide the total kcal/g of each diet

166 – Replace elaborated with a more appropriate word. Could just say HFCD contained SD supplemented with…

170 – Why where # of does unevenly distributed between treatments. Please provide justification for sample size with power calculation.

173 – Please justify that human chorionic gonadotropin would not affect fetal metabolic programming

178 – Just to clarify- HFCD was really fed -8 to -2 week prior to mating, SD is fed for two weeks prior to mating, and HFCD was fed after mating? To me, this is OK, but it must be made much clearer during the description of the treatments.

177 – Please mention explicitly that you there is a companion paper with the same experimental design that measured fertility, body weight, etc. Also, I hope that the body weight presented in Figure 1 is also not reported in the other paper.

194 – Please define that litter is the experimental unit

200 – How were the subsample of pups chosen for measurement?

217 – Replace “breastfeeding” with either “nursing” or “suckling” here and elsewhere.

218 – Replace “produce” with “stimulate”

251 – Be specific about which behavior and physiologic recordings were made

255 – Why was the lactation period only P12 to P31. Shouldn’t it be P1 to P31?

265 – mRNA levels were not determined by extraction. RNA was first extracted and then mRNA expression of UCP1 and CIDEA were determined by RT-qPCR. Correct wording

290 – Duration and intensity of the anticipatory component of are never defined. Please discuss how this was determined because it is completely unclear

288 – You report acrophase, but there is no description of how this is determined. Determination of acrophase must have been performed using cosinor rhythmometry but there is no description of this is the methods. Cosinor rhythmometry should be performed on data where a diurnal response is measured – so for body temp. and for locomotor activity.

292 – Time course data should be analyzed as repeated measures analysis with day as a repeated factor. A two

302 – GraphPad Prism

Results

336 – Replace “ close similarities” with “no difference”

337 – Replace “plasmatic level” with “plasma concentrations” here and elsewhere

Figure 1A – Clarify that this is birthweight within the figure

Figure 5 – Clarify that this is body temperature

Figure 7 – Repsonse variable is not clear from this figure. Figures should be labeled more clearly. Also, it is not explained in the methods how acrophase was determined

Figure 9 – There is no label of which bars represent UCP1 and which represent CIDEA.

Throughout – P-values should be presented both in text and in figures

Discussion

Discussion was not reviewed because of problems with methods and presentation of results.

6. PLOS authors have the option to publish the peer review history of their article (what does this mean?). If published, this will include your full peer review and any attached files.

Reviewer #1: No

Reviewer #2: No

---

## [Author Response · Author response to Decision Letter 0]

25 Feb 2020

Response to Reviewer ♯1

Results:

<< I am not sure Table 1 needs to be reported or could be a supplement, but I will leave that to the editor and author to decide. >>

399 Table 1 was removed from results and included as supplement 1.

<< 741: Access >>

744 the word “asses” was changed to “determine”.

Figures:

<< Resolution of some of the figures is very poor and impossible to read some of the smaller figures. >>

Resolution of images was improved.

Response to Reviewer ♯2

Abstract: 

<< 43- Please more clearly describe the ‘anticipatory component” of body temperature or use different terminology. Same for “energetic components of the rhythm”. The meaning of these phrases is non-obvious >>

45 The sentence “…decrease in the duration and intensity of the anticipatory component, and changes in the most energetic components of the rhythms” was changed to “…decrease in the duration and intensity of the anticipatory rise to nursing, and changes in frequency of the rhythms”.

Introduction: 

<< 117 – Replace with “Women of reproductive age” >>

120 The sentence “…woman in reproductive age” was changed to “…woman of reproductive age”.

<< 120- Be more specific about what conditions these women exhibit >>

123 The sentence “…exhibit some of these conditions” was changed to “…exhibit obesity and metabolic alterations”.

<< 123- Within the introduction there must be justification for why rabbits are used as a model species for maternal overnutrition, and specifically why they can be a good model for humans, since research in human subjects is directly described in the introduction >>

143-160 A justification about the rabbit model was included: “The pathogenesis of human obesity and development of metabolic syndrome (MS) is not fully understood, in order to elucidate the mechanisms and develop new therapeutic strategies, it is essential to have an appropriate animal model that shares the most important aspects of the disease process with humans. Rabbits are a widely used experimental model in biomedical research, and have been proposed as an experimental alternative for the study of MS and its complications, such as atherosclerosis and coronary heart disease, which is the major cause of death in MS patients. Unlike rodents, rabbits have close similarities to human cardiovascular and lipoprotein profiles, with higher levels of apoB-containing low density lipoproteins, and abundant cholesteryl ester transfer protein in plasma, an important regulator of reverse cholesterol transport [Fan et al. 1999; Fan et al. 2015; Fan and Watanabe, 2003; Kawai et al 2006; Furukawa et al 2014; Yin et al. 2002; Such et al 2008; Noujaim et al 2010; Zarzoso et al 2012]. In addition, rabbits fed a high fat and sugar diet develop many characteristics of MS observed in humans [Carroll et al. 1996; Zhang et al 2008]. 

Furthermore, rabbits are an ideal model for the study of transgenerational effects of maternal overnutrition in newborn metabolic regulation, since the placental structure and materno-fetal blood flow interrelationships are closer to the human, in comparison to other models, such as rodents. Humans have discoid and hemochorial type placentas, and the number of trophoblastic layers in the placental barrier, such as the border between fetal and maternal blood systems, differs between species. In humans it is hemomonochorial, with only one layer, while in rabbits it is hemodichorial and in rodents hemotrichorial [Leiser and Kauffman 1994; Review in Perry, 1981; Carter, 2007; Hafez and Tsutsumi, 1966; Furukawa et al.2011]”. 

<< 134 – There is a lot of information in the introduction describing the effects of maternal overnutrition on the circadian clock. However, there is no measurement of the circadian clock gene expression or circadian rhythms of energy metabolism within this paper. There should be a better connection between how the research presented in this paper relates to the circadian clock, or this information should be removed and the introduction should focus more on behavioral and body temperature outputs >>

161 Information associated to the molecular clockwork was reduced in the introduction.

<< 143- A hypothesis must be provided >>

161-167 The hypothesis was modified.

<< 144 – Objective does not include anything about UCP1 or CIDEA expression levels or blood metabolites, so why were these measured? Please add objectives related to these outcomes. Also, circadian rhythms in behavioral and physiologic markers weren’t measured. In order to characterize a true circadian rhythm, responses must be measured in consistent lighting (i.e. 24 dim light). Please refer to the rhythms measured (really only locomoter activity) as a diurnal or daily rhythm. >>

167-171 The objective was modified.

Materials and methods:

<< 156 – Must state the experimental design in the methods >>

183-185 Additional information about the experimental design was included.

<< 164 – Please provide the total kcal/g of each diet >> 

191-193 The total amount Kcal/kg of each diet was included in the text. 

<< 166 – Replace elaborated with a more appropriate word. Could just say HFCD contained SD supplemented with… >>

194 The word “was elaborated” was changed to “contained”.

<< 170 – Why where # of does unevenly distributed between treatments. Please provide justification for sample size with power calculation. >>

199 Due to the difficulties in obtaining the offspring of the HFCD does (important alterations during conception and spontaneous miscarriages) and greater number of dead offspring during the first weeks of life, we decided to increase the number of HFCD does in order to ensure enough animals for the protocol. In this manuscript we did not report any data about female rabbits, such information, in which the power calculation was provided, was submitted elsewhere.

<< 173 – Please justify that human chorionic gonadotropin would not affect fetal metabolic programming >>

202 In our study we administered a single dose (30 UI) of human chorionic gonadotropin (hGC). It is well known that the half-life of this hormone is approximately 37 h �Feiman et al. 1968�, therefore the transitory action of this hormone is short-lived to improve the conception rate. There are no reports in the literature about the effects of a single administration of hGC in fetal metabolic programming. 

<< 178 – Just to clarify- HFCD was really fed -8 to -2 week prior to mating, SD is fed for two weeks prior to mating, and HFCD was fed after mating? To me, this is OK, but it must be made much clearer during the description of the treatments. >>

205-207 Additional information about the feeding procedure was included.

<< 177 – Please mention explicitly that you there is a companion paper with the same experimental design that measured fertility, body weight, etc. Also, I hope that the body weight presented in Figure 1 is also not reported in the other paper. >>

203 The information was removed, since this is part of another manuscript submitted elsewhere. 

<< 194 – Please define that litter is the experimental unit >>

224 With the term litter, we refer to “the offspring at one birth of a multiparous animal” �Webster's Dictionary of The English Language�.

<< 200 – How were the subsample of pups chosen for measurement? >>

230 The word “randomly” was included in the sentence. 

<< 217 – Replace “breastfeeding” with either “nursing” or “suckling” here and elsewhere. >>

247 The word “nursing” was included in the sentence.

<< 218 – Replace “produce” with “stimulate” >>

248 The word “stimulate” was included in the sentence.

<< 251 – Be specific about which behavior and physiologic recordings were made >>

281 In the sentence, the term “core body temperature and gross locomotor activity” was included.

<< 255 – Why was the lactation period only P12 to P31. Shouldn’t it be P1 to P31? >>

283-284 Due to the size of the newborn rabbit, the surgery for transponder implantation was made at postnatal day 8 and four days were considered for the complete surgery recovery of the pups. At this time, the telemetry recordings were initiated. 

<< 265 – mRNA levels were not determined by extraction. RNA was first extracted and then mRNA expression of UCP1 and CIDEA were determined by RT-qPCR. Correct wording >>

289 The sentence was corrected.

<< 290 – Duration and intensity of the anticipatory component of are never defined. Please discuss how this was determined because it is completely unclear >>

339-346 Information about the duration and intensity of the anticipatory component was included.

<< 288 – You report acrophase, but there is no description of how this is determined. Determination of acrophase must have been performed using cosinor rhythmometry but there is no description of this is the methods. Cosinor rhythmometry should be performed on data where a diurnal response is measured – so for body temp. and for locomotor activity. >>

Originally, in the manuscript we indicate that “To evaluate the rhythmicity in the rabbits’ core body temperature and locomotor activity, we used a previously reported procedure �Montúfar-Chaveznava R, et al. 2013; Trejo-Muñoz L, et al. 2012�”. This was replaced by a full description of the rhythmicity procedure (332-339).

It is well known that single Cosinor was developed to analyze short and sparse data series [Halberg et al., 1967]. In this work, as we have a long time series for body temperature and locomotor activity, we employed two different methods. The first method used was the Fourier theory, to determine the amplitude and phase (or acrophase) of specific sinusoids. In particular, we employed the Fast Fourier Transform for frequency decomposition considering we have discrete data; next we selected the frequencies (or periods) of interest (i.e. 24h) and recovered the information from the polar representation of Fourier coefficients. In the second method used, since the 24 h phase obtained with Fast Fourier Transform does not always fit the data, we constructed a pulse sequence of a 24h period which was shifted along the data to find time pulse and data fits using a similarity metric.

<< 292 – Time course data should be analyzed as repeated measures analysis with day as a repeated factor. A two >>

The data was analyzed as repeated measures, as mentioned.

<< 302 – GraphPad Prism >>

365 The word “GraphaPad” was corrected to “GraphPad Prism”

Results:

<< 336 – Replace “ close similarities” with “no difference” >>

399 The term “close similarities” was replaced with “no difference”.

<< 337 – Replace “plasmatic level” with “plasma concentrations” here and elsewhere >>

400 The term “plasmatic levels” was replaced with “plasma concentrations”.

Figures:

All the figures were modified according to comments.

References:

Carroll JF, Dwyer TM, Grady AW, Reinhart GA, Montani JP, Cockrell K, et al. Hypertension, cardiac hypertrophy, and neurohumoral activity in a new animal model of obesity. Am J Physiol. 1996;271: H373-H378.

Carter AM. Animal models of human placentation--a review. Placenta. 2007;28 (Suppl A): S41-S47. 

Fan J, Araki M, Wu L, Challah M, Shimoyamada H, Lawn RM, et al. Assembly of lipoprotein (a) in transgenic rabbits expressing human apolipoprotein (a). Biochem Biophys Res Commun. 1999;255: 639-44.

Fan J, Kitajima S, Watanabe T, Xu J, Zhang J, Liu E, et al. Rabbit models for the study of human atherosclerosis: from pathophysiological mechanisms to translational medicine. Pharmacol Ther. 2015;146: 104-119. doi: 10.1016/j.pharmthera.2014.09.009.

Fan J, Watanabe T. Transgenic rabbits as therapeutic protein bioreactors and human disease models. Pharmacol Ther. 2003;99: 261-282. 

Furukawa S, Kuroda Y, Sugiyama A. J A comparison of the histological structure of the placenta in experimental animals. Toxicol Pathol. 2014;27: 11-18. doi:10.1293/tox.2013-0060. 

Hafez ES, Tsutsumi Y. Changes in endometrial vascularity during implantation and pregnancy in the rabbit. Am J Anat. 1966;118: 249-282. 

Halberg F, Tong YL, Johnson EA: Circadian System Phase – An Aspect of Temporal morphology; Procedures and Illustrative Examples. In Proc. International Congress of Anatomists. The Cellular Aspects of Biorhythms, Symposium on Biorhythms. Edited by Mayersbach HV. New York: Springer-Verlag; 1967:20-48.

Kawai T, Ito T, Ohwada K, Mera Y, Matsushita M, Tomoike H. Hereditary postprandial hypertriglyceridemic rabbit exhibits insulin resistance and central obesity: a novel model of metabolic syndrome. Arterioscler Thromb Vasc Biol. 2006;26: 2752-2757.

Leiser R, Kaufmann P. Placental structure: in a comparative aspect. Exp Clin Endocrinol. 1994;102: 122-134.

Montúfar-Chaveznava R, Trejo-Muñoz L, Hernández-Campos O, Navarrete E, Caldelas I. Maternal olfactory cues synchronize the circadian system of artificially raised newborn rabbits. PLoS One. 2013;8: e74048. doi: 10.1371/journal.pone.0074048. eCollection 2013.

Noujaim SF, Stuckey JA, Ponce-Balbuena D, Ferrer-Villada T, López-Izquierdo A, Pandit S, et al. Specific residues of the cytoplasmic domains of cardiac inward rectifier potassium channels are effective antifibrillatory targets. FASEB J. 2010;24: 4302-4312. doi: 10.1096/fj.10-163246. 

Perry JS The mammalian fetal membranes. J Reprod Fertil. 1981;62: 321-335.

Such L, Alberola AM, Such-Miquel L, López L, Trapero I, Pelechano F, et al. Effects of chronic exercise on myocardial refractoriness: a study on isolated rabbit heart. Acta Physiol (Oxf). 2008;193: 331-339. doi: 10.1111/j.1748-1716.2008.01851.x. 

Trejo-Muñoz L, Navarrete E, Montúfar-Chaveznava R, Caldelas I. Determining the period, phase and anticipatory component of activity and temperature patterns in newborn rabbits that were maintained under a daily nursing schedule and fasting conditions. Physiol Behav. 2012;106: 587-596. doi: 10.1016/j.physbeh.2012.04.005.

Yin W, Yuan Z, Wang Z, Yang B, Yang Y. A diet high in saturated fat and sucrose alters glucoregulation and induces aortic fatty streaks in New Zealand White rabbits. Int J Exp Diabetes Res. 2002;3: 179-184.

Zarzoso M, Such-Miquel L, Parra G, Brines-Ferrando L, Such L, Chorro FJ, et al. The training-induced changes on automatism, conduction and myocardial refractoriness are not mediated by parasympathetic postganglionic neurons activity. Eur J Appl Physiol. 2012;112: 2185-2193. doi: 10.1007/s00421-011-2189-4. 

Zhang XJ, Chinkes DL, Aarsland A, Herndon DN, Wolfe RR. Lipid metabolism in diet-induced obese rabbits is similar to that of obese humans. J Nutr. 2008;138: 515-518.

---

## [Editor Report · Decision Letter 1]

15 Apr 2020

Misadjustment of diurnal expression of core temperature and locomotor activity in lactating rabbits associated with maternal over-nutrition before and during pregnancy

PONE-D-19-28260R1

Dear Dr. Ivette Caldelas,

We are pleased to inform you that your manuscript has been judged scientifically suitable for publication and will be formally accepted for publication once it complies with all outstanding technical requirements.

With kind regards,

De-Hua Wang, Ph.D.

Academic Editor

PLOS ONE

Additional Editor Comments (optional):

I have gone through the comments from the reviewers and the responses from the authors. I think authors have revised their ms according the comments and suggestion from the editor and reviewers.

I recommend this ms can be accepted for publication in Plos One.
---

## [Editor Report · Acceptance letter]

28 Apr 2020

PONE-D-19-28260R1 

Misadjustment of diurnal expression of core temperature and locomotor activity in lactating rabbits associated with maternal over-nutrition before and during pregnancy 

Dear Dr. Caldelas:

I am pleased to inform you that your manuscript has been deemed suitable for publication in PLOS ONE. Congratulations! Your manuscript is now with our production department. 

With kind regards,

on behalf of

Prof. De-Hua Wang 

Academic Editor

PLOS ONE